# MAXIMUM COVERAGE IN TURNSTILE STREAMS WITH APPLICATIONS TO FINGERPRINTING MEASURES

## ABSTRACT

In the maximum coverage problem we are given $d$ subsets from a universe $[n]$, and the goal is to output at most $k$ subsets such that their union covers the largest possible number of distinct items. The input can be formalized as an $n \times d$ matrix $\boldsymbol{A}$ where entry $\boldsymbol{A}_{ij} \neq 0$ if item $i$ is covered by subset $j$ and $\boldsymbol{A}_{ij} = 0$ otherwise. In this paper we create the first linear sketch to solve the maximum coverage problem. The sketch has size sublinear in the input and is directly applicable to distributed and streaming settings, often offering significant runtime improvements. We focus on the application to the turnstile streaming model which supports insertions and *deletions*. In this model, updates take the form $(i, j, \pm 1)$ which update $\boldsymbol{A}_{ij}$ to $\boldsymbol{A}_{ij} + 1$ or $\boldsymbol{A}_{ij} - 1$, depending on the sign. Previous work has largely focused on more restrictive models, such as the set-arrival model where each update reveals an entire column of $\boldsymbol{A}$, or the insertion-only model which does not allow deletions. We design an algorithm with an $\tilde{O}(d/\varepsilon^3)$ space bound for all $k \geq 0$. We note that when $k$ is constant, this space bound is nearly optimal up to logarithmic factors. We then turn to fingerprinting for risk measurement. The input is an $n \times d$ matrix $\boldsymbol{A}$ where there are $n$ users and $d$ features, and the goal is to determine which $k$ features (or columns in $\boldsymbol{A}$) together pose the greatest re-identification risk. Our maximum coverage sketch directly enables a solution to targeted fingerprinting for risk measurement. Furthermore, we present a result of independent interest: a linear sketch of the complement of $F_p$, the $p^{\text{th}}$ frequency moment, for $p \geq 2$. We use this sketch to solve general fingerprinting for risk management. Empirical evaluation confirms the practicality of our fingerprinting algorithms, demonstrating a speedup of up to 210x over prior work. We also demonstrate that our general fingerprinting algorithm can serve as a dimensionality reduction technique, with an application to facilitating enhanced feature selection efficiency.

## 1 INTRODUCTION

Maximum coverage is a classic NP-hard problem with applications including information retrieval (Anagnostopoulos et al., 2015), influence maximization (Kempe et al., 2003), and sensor placement (Krause & Guestrin, 2007). Given $d$ subsets of a universe with $n$ items and cardinality constraint $k \geq 0$, the goal is to output the $k$ subsets whose union covers the greatest number of distinct items. A simple greedy algorithm solves this problem by running for $k$ rounds, selecting the subset with the largest marginal gain in each round. This algorithm, in polynomial time and space, achieves a $1 - 1/e$ relative approximation, an approximation factor which is tight for polynomial time algorithms unless $P = NP$ (Feige, 1998). However, its polynomial time and space complexity make it impractical for handling massive datasets. Our objective, consequently, is to study algorithms for maximum coverage with *sublinear* time and memory requirements.

We formalize the input to the maximum coverage problem as an $n \times d$ matrix $\boldsymbol{A}$ where entry $\boldsymbol{A}_{ij}$ is nonzero if item $i$ is in subset $j$ and 0 otherwise. In this paper we create the first *linear* sketch with size sublinear in the input matrix to solve maximum coverage, to the best of our knowledge. Linear sketches compress large input matrices while preserving essential information used to form the final output. They also support updates, including both insertions and deletions, to the input matrix. In the context of maximum coverage, an update takes the form $(i, j, \pm 1)$ which modifies entry $\boldsymbol{A}_{ij}$ by adding or subtracting one, effectively adding or removing an item from a subset (or doing nothing

if entry $\boldsymbol{A}_{ij}$ was nonzero before and after the update). After all updates, we can query the sketch to form our output (in this case to select the $k$ subsets to output).

Linear sketches are far more powerful than algorithms tailored to specific models, as they enable significant runtime improvements while being applicable to a wide range of settings including distributed and streaming contexts[1]. We focus on the application to the turnstile streaming setting where updates come one-by-one in a stream and each update modifies $\boldsymbol{A}_{ij}$ by adding or subtracting one. To our knowledge, our linear sketch provides the first streaming algorithm for maximum coverage which allows arbitrary deletions of items from subsets. Deletions are critical for a number of applications. For example, we use them to extend our algorithm to fingerprinting for dataset risk measurement.

**Related Work.** There is an extensive body of work on the maximum coverage problem, and we do not attempt to give a comprehensive overview here. Instead, we focus on the related streaming literature. Specifically, we discuss one-pass streaming algorithms with polynomial time complexity, where a pass refers to a single traversal of the stream of updates. In the following, a $(x)$ relative approximation means that the number of distinct items covered by the $k$ subsets selected by the algorithm is at least $x \cdot \textbf{OPT}$, where $\textbf{OPT}$ denotes the number of items covered by the optimal solution. In addition, $\tilde{O}(\cdot)$ notation is used to suppress poly-logarithmic factors in its argument.

McGregor & Vu (2018) provide a one-pass algorithm that outputs a $(1-1/e-\varepsilon)$ relative approximation for $\varepsilon \in (0,1)$ in $\tilde{O}(d/\varepsilon^2)$ space. They consider the insertion-only set-arrival streaming model which given our input $n \times d$ matrix $\boldsymbol{A}$ is equivalent to seeing an entire column of $\boldsymbol{A}$ in each update. In other words, each update reveals a subset and the items it covers, and deletions are not supported. At a high level, their algorithm first subsamples rows of $\boldsymbol{A}$ such that $\textbf{OPT}$ in this smaller universe is $\tilde{O}(k/\varepsilon^2)$. They then argue that achieving a $(1 - 1/e)$ relative approximation to maximum coverage on this smaller universe achieves a $(1 - 1/e - \varepsilon)$ relative approximation overall.

Bateni et al. (2017) give a one-pass $(1-1/e-\varepsilon)$ relative approximation algorithm that uses $\tilde{O}(d/\varepsilon^3)$ memory. They consider the insertion-only streaming model which given our input matrix $\boldsymbol{A}$ is equivalent to receiving updates of the form $(i, j, 1)$. Note that negative updates (i.e., deletions) are not supported. They specifically provide an algorithm that carefully samples a number of nonzero entries of input $\boldsymbol{A}$, and they show that any $(\alpha)$ relative approximation on this smaller subsampled universe achieves an $(\alpha - \varepsilon)$ relative approximation for the original input. We use their sketch as a starting point (see Section 3 for details).

There has also been work that achieves different approximation factors (Saha & Getoor, 2009; McGregor et al., 2021), in random-arrival streams (Warneke et al., 2023; Chakrabarti et al., 2024), and in more general submodular maximization in the insertion-only set-arrival model (Badanidiyuru et al., 2014; Kazemi et al., 2019).

In contrast to all of the above, our sketch (and therefore turnstile streaming algorithm) allows deletions *and* arbitrarily ordered updates to any individual entry of $\boldsymbol{A}$.

We note that there has also been work on submodular maximization in the somewhat related dynamic model (Monemizadeh, 2020; Chen & Peng, 2022; Lattanzi et al., 2020). We briefly outline the differences between the dynamic model and streaming model. While both models process updates sequentially, the key distinction lies in their primary objectives. The dynamic model prioritizes achieving minimal update time, whereas the streaming model, which we consider here, emphasizes minimizing space usage. It is worth noting that most algorithms designed for the dynamic model do not achieve sublinear space and, in some cases, require exponential space. Despite our emphasis on space efficiency, the linear sketches we present that we will apply to the streaming model also maintain sublinear update times.

**Fingerprinting for Risk Management.** We also design linear sketches which extend to turnstile streaming algorithms for targeted and general fingerprinting for risk management, achieving approximation factors which are near-optimal for polynomial-time algorithms unless $P = NP$ (Gulyás et al., 2016). In targeted fingerprinting, the input is an $n \times d$ matrix $\boldsymbol{A}$, where $n$ represents the number of users and $d$ represents the number of features, and a target user $u \in [n] = \{1, 2, \ldots, n\}$. The value of entry $\boldsymbol{A}_{ij}$ denotes the value of the user $i$ at feature $j$. The goal is to identify at most

---

[1]Refer to Section 2 for more details on linear sketches.

$k$ features $\{f_1, f_2, \ldots, f_k\}$ such that the number of users who share identical values to the target user $u$ at these positions $\{f_1, f_2, \ldots, f_k\}$ is minimized. In general fingerprinting, the input is also an $n \times d$ matrix $\boldsymbol{A}$ where $n$ is the number of users and $d$ is the number of features. Here, the goal is to identify at most $k$ features $\{f_1, f_2, \ldots, f_k\}$ such that number of pairs of users who share identical values at these positions $\{f_1, f_2, \ldots, f_k\}$ is minimized.

Our algorithms fit into the broader privacy attack literature (Seonghun et al., 2023; Chia et al., 2019; Zhou et al., 2023) and can be seen as an extension of Chia et al. (2019) in the area of privacy auditing and risk measurement. Specifically, these algorithms address the issue of fingerprinting, a technique used to re-identify users from datasets, which poses a significant privacy risk. Fingerprinting refers to the process of identifying a user based on unique combinations of attributes (or feature values) in a dataset. Our algorithms help mitigate this risk by identifying which $k$ features in a dataset are most likely to enable adversaries to successfully fingerprint users to prioritize data protection. Previous work outside of Gulyás et al. (2016), whose linear space and time algorithms we improve upon, has only measured the risk of a whole dataset or a fixed set of features. In contrast, our time and space efficient algorithms are suitable for real-time monitoring and continuous measure of re-identification risks even as the dataset changes over time. In addition, targeted fingerprinting is a form of frequency estimation and could be useful in other contexts such as discovering heavy hitters (Bhattacharyya et al., 2016; Zhu et al., 2020).

## 1.1 OUR CONTRIBUTIONS

In all of the following $\tilde{O}(\cdot)$ suppresses logarithmic factors in its argument. The update time of a sketch refers to the time required to update and maintain the sketch after an update, and the reporting time refers to the time required to return the result upon a query.

**Maximum Coverage Results.**

**Theorem 1.** *There exists a linear sketch of size $\tilde{O}(d/\varepsilon^3)$ with update time $\tilde{O}(d/\varepsilon^3)$ and reporting time $\tilde{O}(kd/\varepsilon^3)$ such that given $d$ subsets of a universe $[n]$, integer $k \geq 0$, and $\varepsilon \in (0,1)$, running an $(1 - 1/e)$ relative approximation algorithm on the sketch produces a $(1 - 1/e - \varepsilon)$ relative approximate solution the to maximum coverage problem with probability at least $1 - 1/\text{poly}(d)$.*

Note that the only dependence on $k$ in our space complexity (despite our algorithm working for all $k \geq 0$) appears in poly-logarithmic factors. Moreover, since we can assume $k \leq d$ - otherwise, we can simply output all the input subsets - these poly-logarithmic factors in $k$ can be treated as poly-logarithmic factors in $d$, which are hidden by the $\tilde{O}(\cdot)$ notation.

Our linear sketch is then naturally applicable to the turnstile streaming model since linear sketches accomodate insertions and deletions.

**Corollary 1.1.** *Given $d$ subsets of a universe $[n]$, integer $k \geq 0$, and $\varepsilon \in (0,1)$, there exists a one-pass turnstile streaming algorithm that with probability at least $1 - 1/d$ gives a near-optimal $(1 - 1/e - \varepsilon)$ relative approximation to maximum coverage in $\tilde{O}(d/\varepsilon^3)$ space.*

We note that the space complexity of our algorithm matches that of Bateni et al. (2017) and, for constant $\varepsilon$, that of McGregor & Vu (2018). Additionally, several lower bounds exist.

Assadi (2017) shows that achieving a $(1 - \varepsilon)$ relative approximation in a constant number of passes requires $\Omega(d/\varepsilon^2)$ space. Assadi & Khanna (2018) shows that even achieving a $n^{1/3}$ or $\sqrt{k}$ relative approximation in one pass with a sketch requires the sketch to have size $\Omega(d/k^2)$. McGregor & Vu (2018) shows that achieving better than a $1 - 1/e$ approximation in a constant number of passes requires $\Omega(d/k^2)$ space. Therefore (while our algorithm works for all $k \geq 0$ with space $\tilde{O}(d/\varepsilon^3)$), if $k$ is constant, our result is optimal up to poly-logarithmic factors. Bateni et al. (2017) also show that any $(1/2 + \varepsilon)$ relative approximation multi-pass streaming algorithm requires $\Omega(d)$ space.

**Fingerprinting Results.** We then use our linear sketch from Theorem 1 to create a linear sketch to solve targeted fingerprinting for risk management, improving upon the linear time and space algorithm of Gulyás et al. (2016). To reduce targeted fingerprinting to maximum coverage, we subtract the value of entry $\boldsymbol{A}_{uj}$ from each $\boldsymbol{A}_{ij}$ for all $i \in [n], j \in [d]$. Recall that $u$ is the input "target" user. This reduction is feasible only because our maximum coverage sketch accommodates deletions. Here, a $(x)$ relative approximation means that the number of users separated from the

input user $u$ by the $k$ features selected by the algorithm is at least $x \cdot$ **OPT**, where **OPT** denotes the number of users separated from $u$ by the optimal solution. The proof of the following is deferred to Appendix A.3.

**Corollary 1.2.** *Given $n \times d$ matrix $\boldsymbol{A}$, target user $u \in [n]$, and $\varepsilon \in (0, 1)$, there exists a linear sketch of size $\tilde{O}(d/\varepsilon^3)$ with update time $\tilde{O}(d/\varepsilon^3)$ and reporting time $\tilde{O}(kd/\varepsilon^3)$ such that running a $(1 - 1/e)$ relative approximation algorithm on the sketch produces a $(1 - 1/e - \varepsilon)$ relative approximate solution to targeted fingerprinting with probability at least $1 - 1/\text{poly}(d)$.*

This is again directly applicable to the turnstile streaming model.

**Corollary 1.3.** *Given $n \times d$ matrix $\boldsymbol{A}$, target user $u \in [n]$, and $\varepsilon \in (0, 1)$, there exists a one-pass turnstile streaming algorithm that achieves a $(1 - 1/e - \varepsilon)$ relative approximation to targeted fingerprinting using space $\tilde{O}(d/\varepsilon^3)$ with probability at least $1 - 1/d$.*

We also improve upon the linear time and space algorithm of Gulyás et al. (2016) for general fingerprinting for risk management. However, unlike targeted fingerprinting, reducing general fingerprinting to maximum coverage (as Gulyás et al. (2016) does) requires tracking, for all $\binom{n}{2}$ pairs of users, whether they differ in value on a certain feature. This results in a $O(n^2) \times d$ input matrix, making it infeasible to handle updates with linear sketches, which we use to accommodate deletions. Upon receiving an update to some entry of $\boldsymbol{A}$, the sketch must be updated for all pairs of users that are either newly separated or no longer separated by a given feature. This could involve updating all $O(n^2)$ pairs. Therefore, we design an algorithm for general fingerprinting with a near-optimal $(1 - 1/e - \varepsilon)$ relative approximation in a different way.

To do this, we first present a framework for submodular maximization under cardinality constraints over monotone, linearly sketchable functions in turnstile streams [2]. Submodular functions exhibit the property of diminishing returns, and we specifically focus on maximizing monotone, non-negative submodular functions that are defined over subsets of a given universe. In our context, this means that there are $d$ subsets of a universe $[n]$, and the function takes as input some of these subsets and returns a positive real number. A function is defined to be linearly sketchable if its input can be compressed by a linear sketch and this sketch can be queried to efficiently produce the function's output value on some given subsets. For formal definitions of submodular functions and linearly sketchable functions, see Appendix A.1.2 and Appendix A.1.2. Here, a $(x)$ relative approximation means that the output of the function on the $k$ subsets selected by the algorithm is at least $x \cdot$ **OPT**, where **OPT** denotes the maximum output of the function on $k$ subsets. The proof of the following is deferred to Appendix A.4.

**Theorem 2.** *Given $d$ subsets of a universe $[n]$ and $\varepsilon \in (0, 1)$, take $f$ to be a submodular, monotone, non-negative function over subsets that we want to maximize by selecting at most $k$ subsets. If $f$ is linearly sketchable with a $(1 \pm \gamma)$ relative approximation in $O(s)$ space, if we set $\gamma = \varepsilon/k$, then there exists an one-pass turnstile streaming algorithm that outputs a $(1 - 1/e - \varepsilon)$ relative approximation using $O(sk)$ space. The algorithm succeeds with probability at least $1 - 1/n$ assuming that querying the sketch results in error at most $O(1/(ndk))$.*

We then instantiate this framework to solve general fingerprinting. To do this, we design a novel sketch for estimating the quantity $n^p - F_p$ for $p \geq 2$ where $F_p$ is the $p^{\text{th}}$ frequency moment. Here, we are given a $n$-dimensional vector $\mathbf{x}$, $\mathcal{Z}$ is the set of distinct values in vector $\mathbf{x}$, and $f_i$ is the frequency of the $i^{\text{th}}$ distinct value in $\mathbf{x}$. For example, take $\mathbf{x} = (1, 5, 5, 3, -2, 3, 3, 7, 3)$. Here the distinct values are $1, 5, 3, -2$, and $7$ and the respective frequencies of those values are $1, 2, 4, 1$, and $1$. So $F_p = \sum_{i \in \mathcal{Z}} f_i^p = 1^p + 2^p + 4^p + 1^p + 1^p$. The quantity $n^p - F_p$ intuitively counts the number of $p$-tuples that can be formed from the entries of $\mathbf{x}$ (with repetition) where not all entries of the tuple are identical in value. Here, updates are of the form $(i, \pm 1)$ which performs $x_i \leftarrow x_i \pm 1$.

**Theorem 3.** *There exists a linear sketch of size $\tilde{O}(\gamma^{-2})$ with update time $\tilde{O}(\gamma^{-2})$ and reporting time $\tilde{O}(\gamma^{-2})$ that given a $n$-dimensional vector $\mathbf{x}$, constant integer $p \geq 2$, and $\gamma, \delta \in (0, 1)$ outputs a $(1 \pm \gamma)$ relative approximation of $n^p - F_p$ with probability at least $1 - \delta$.*

We believe this sketch to be of independent interest since it is of the *complement* of the frequency moment of a dataset. The $p^{\text{th}}$ frequency moment, denoted as $F_p$, is computed by taking the frequency

---

[2]Linear sketching is applicable to a wide variety of functions in different contexts including regression, low rank approximation and graph compression, see, e.g., Woodruff (2014).

of each distinct item, raising it to the $p^{\text{th}}$ power, and summing the results. Frequency moments have numerous applications. For example, $F_p$ for $p \geq 2$ can indicate the degree of the skew of data which is used in the selection of algorithms for data partitioning (Dewitt et al., 2000), error estimation (Ioannidis & Poosala, 1995), and more. See Alon et al. (1999) for a more in-depth discussion. There are also direct applications for the quantity $n^p - F_p$ such as our use of the sketch to solve general fingerprinting. The proof of the following is deferred to Appendix A.6.

**Theorem 4.** *There exists a linear sketch of size $\tilde{O}(dk^3/\varepsilon^2)$ with update and reporting time $\tilde{O}(dk^3/\varepsilon^2)$ that, given $d$ subsets of a universe $[n]$ and $\varepsilon \in (0,1)$, outputs with probability at least $1 - 1/n$ a near-optimal $(1 - 1/e - \varepsilon)$ relative approximation to general fingerprinting.*

**Corollary 1.4.** *Given $d$ subsets of a universe $[n]$ and $\varepsilon \in (0,1)$, there exists an one-pass turnstile streaming algorithm which outputs with probability at least $1 - 1/n$ a near-optimal $(1 - 1/e - \varepsilon)$ approximation to general fingerprinting in space $\tilde{O}(dk^3/\varepsilon^2)$.*

**Experimental Results.** We also illustrate the practicality of our fingerprinting algorithms by running experiments on two different datasets of size $32{,}000 \times 80$ and $2{,}500{,}000 \times 120$. In a direct comparison with the implementations of Gulyás et al. (2016), our algorithms show significantly improved efficiency while retaining high comparative accuracy. Specifically, for targeted fingerprinting, we achieve a speedup of up to $49$x, with accuracy that converges rapidly to that of Gulyás et al. (2016). For general fingerprinting, we gain a speedup of up to 210x while again achieving high comparative accuracy.

Finally, we believe that our general fingerprinting algorithm can serve as a dimensionality reduction technique. To illustrate this, we apply it in the context of feature selection for machine learning models where feature spaces are often extremely large. Feature selection is a process that identifies a subset of relevant features from the original dataset to improve model performance or computational efficiency. By using our general fingerprinting algorithm to perform feature selection and therefore reduce the dimensionality of the input dataset, we can mitigate issues such as overfitting, improve interpretability, and greatly speed up machine learning algorithms.

In particular, since the time complexity of many popular clustering algorithms such as $k$-means scales with the dimensionality of the data, we use our general fingerprinting algorithm to select $x$ features that best separate the data. We therefore have reduced the dimension of the feature space to $x$. We then use $k$-means on these $x$ features instead of the full feature space and demonstrate that this approach significantly increases efficiency while sacrificing little in terms of accuracy. We believe our techniques to be general and extendable to other clustering and machine learning algorithms outside of $k$-means.

## 2 Preliminaries

**Notation.** Some preliminaries are postponed to Appendix A.1. We denote $\boldsymbol{A}_{ij}$ as the entry at the $i^{\text{th}}$ row and $j^{\text{th}}$ column of matrix $\boldsymbol{A}$. $\tilde{O}(\cdot)$ notation suppresses logarithmic factors in its argument.

**Linear Sketches** We begin by defining what a linear sketch is and then provide an overview of the specific linear sketches used in this paper. Given a $n \times d$ matrix $\boldsymbol{A}$, we can compress it while retaining essential information to solve the problem by multiplying it with a $r \times n$ linear sketching matrix $\boldsymbol{S}$. A linear sketch is a matrix drawn from a certain family of random matrices independent of $\boldsymbol{A}$. This independence ensures that $\boldsymbol{S}$ can be generated without prior knowledge of the contents of $\boldsymbol{A}$. Linear sketches support insertions and deletions to the entries of $\boldsymbol{A}$, as $\boldsymbol{S}(\boldsymbol{A} + c_{ij}) = \boldsymbol{S}\boldsymbol{A} + \boldsymbol{S}c_{ij}$ holds for any update $c_{ij}$, which adds or subtracts one from an entry of $\boldsymbol{A}$. This property allows us to maintain $\boldsymbol{S}\boldsymbol{A}$ throughout updates without requiring storage of $\boldsymbol{A}$ itself. Furthermore, $\boldsymbol{S}$ is typically stored in an implicit, pseudorandom form (e.g., via hash functions) rather than explicitly, enabling efficient sketching of updates $c_{ij}$. The primary focus is on minimizing the space requirement of a linear sketch, specifically ensuring that the sketching dimension $r$ is sublinear in $n$ and ideally much smaller. Alongside space efficiency, there are two additional important performance metrics: update time and reporting time. Update time refers to the time complexity required for the sketch to process an update, and reporting time refers to the time complexity needed to return an answer to a query.

**Perfect $L_0$ Sampling.** Consider an underlying vector $\mathbf{x} = (\mathbf{x}_1, \mathbf{x}_2, \ldots, \mathbf{x}_n)$. Let $\text{Supp}(\mathbf{x})$ be the set of nonzero elements of $\mathbf{x}$. A perfect $L_0$ sampler, with probability $1 - \delta$, returns a tuple $(i, \mathbf{x}_i)$ for

$\mathbf{x}_i \in \text{Supp}(\mathbf{x})$ such that $\Pr[i = j] = \frac{1}{\|\mathbf{x}\|_0} \pm n^{-c}$ for every $\mathbf{x}_j \in \text{Supp}(\mathbf{x})$ for large constant $c$. Note that it returns the value of $\mathbf{x}_i$ exactly with no error. With probability $\delta$, the sampler outputs FAIL. An $L_0$ sampler can be seen as a linear sketch and accommodates both insertions and deletions to the underlying vector $\mathbf{x}$. The parameter $n^{-c}$ can be made arbitrarily small by increasing constant $c$, effectively making the sampling process indistinguishable from perfect uniform random sampling of nonzero entries. Importantly, increasing $c$ incurs only constant factors in space usage. Jowhari et al. (2010) give an algorithm that achieves this in $O(\log^2 n \log(1/\delta))$ bits of space. By inspecting Theorem 2 of Jowhari et al. (2010) and using appropriate sparse recovery schemes we can see that the update and reporting time are both $\text{poly}(\log n) \cdot \log(1/\delta))$.

$L_0$ **Sketch.** Consider an underlying vector $\mathbf{x} = (\mathbf{x}_1, \dots, \mathbf{x}_n)$ where all entries are initially set to 0. We receive $m$ updates of the form $(i, v) \in [n] \times \{-M, \dots, M\}$ in a stream where the update performs $\mathbf{x}_i \leftarrow \mathbf{x}_i + v$. At the end of the stream, the goal is to output a $(1 \pm \varepsilon)$ relative approximation of $L_0$ with probability at least $1 - \delta$ where $L_0 = |\{i : \mathbf{x}_i \neq 0\}|$. Kane et al. (2010) give a $L_0$ sketch with $O(1)$ update and reporting time that requires $O(\epsilon^{-2} \log n(\log(1/\epsilon) + \log \log(mM)) \cdot \log(1/\delta))$ memory. A $L_0$ sketch is a linear sketch and accommodates both insertions and deletions to the underlying vector $\mathbf{x}$.

**Moment Estimation.** Consider an underlying vector $\mathbf{x} = (\mathbf{x}_1, \mathbf{x}_2, \dots, \mathbf{x}_n)$. For all $i \in [n]$, $\mathbf{x}_i \in [m]$. Let $f_i = |\{j : \mathbf{x}_j = i\}|$ be the number of occurrences of value $i$ in $\mathbf{x}$. We define the $p^{\text{th}}$ frequency moment of $\mathbf{x}$ as $F_p \stackrel{\text{def}}{=} \sum_{i=1}^m f_i^p$ for $p \geq 0$.

## 3 MAX-COVERAGE ALGORITHM

We now present our sketch, Max-Coverage-LS (Algorithm 5), to prove Theorem 1. The proofs are deferred to Appendix A.2. Recall that the input is formalized as a $n \times d$ matrix $\boldsymbol{A}$, where entry $\boldsymbol{A}_{ij}$ is nonzero if $i$ is in subset $j$, and 0 otherwise. Our approach uses Algorithm 1 from Bateni et al. (2017) as a starting point. Bateni et al. (2017) reduce the original input matrix $\boldsymbol{A}$ to a smaller universe $\boldsymbol{A}_*$ by carefully sampling a subset of its nonzero entries. They then show that running the greedy algorithm on this smaller universe yields a $(1 - 1/e - \varepsilon)$ relative approximation for the maximum coverage problem on $\boldsymbol{A}$.

The plan for this section is as follows. Initially, we will not consider the streaming setting; instead, we will assume the standard RAM model, where the entire input matrix $\boldsymbol{A}$ is fully accessible. We will first introduce the smaller universe $\boldsymbol{A}_*$, describing its properties and role in the problem. Next, we will show how to construct $\boldsymbol{A}_*$ within the RAM model, ensuring that the construction is easily adapted to handle updates efficiently. Finally, we will present our complete algorithm, a linear sketch, and detail how it enables the construction of $\boldsymbol{A}_*$ in a manner that accommodates insertions and deletions.

Constructing $\boldsymbol{A}_*$ involves permuting the items (rows) of $\boldsymbol{A}$ and processing them in the order determined by the permutation. For each item (row) $i$, a subset of $\tilde{O}(d/(\varepsilon k))$ nonzero entries from the $i^{\text{th}}$ row of $\boldsymbol{A}$ is arbitrarily selected and added to $\boldsymbol{A}_*$. This process continues until $\boldsymbol{A}$ contains $\tilde{O}(d/\varepsilon^3)$ nonzero entries in total. $\boldsymbol{A}^*$ is a carefully subsampled version of $\boldsymbol{A}$, where only $\tilde{O}(d/\varepsilon^3)$ of the nonzero entries are retained while the rest are set to 0. We restate their algorithm $\boldsymbol{A}_*(k, \varepsilon, \delta)$ (Algorithm 3) in Appendix A.2. In Bateni et al. (2017) this subsampled matrix is referred to as $H_{\leq d}$.

The authors of Bateni et al. (2017) prove that solving the maximum coverage problem on $\boldsymbol{A}_*(k, \varepsilon, \delta)$ with a $\alpha$-relative approximation guarantees a $(\alpha - \varepsilon)$-relative approximation on the original matrix $\boldsymbol{A}$ with high probability. The final $(1 - 1/e - \varepsilon)$-relative approximation is achieved using k-cover (Algorithm 4), which sets appropriate parameters and applies the greedy algorithm (or any $(1 - 1/e)$ approximation algorithm) to $\boldsymbol{A}_*$.

**Theorem 5** (Theorem 2.7 and 3.1 of Bateni et al. (2017)). *Running k-cover with $\boldsymbol{A}_*$ produces a $(1 - 1/e - \varepsilon)$ approximate solution to maximum coverage with probability $1 - 1/d$.*

We now show how to build our linear sketch. First, we will specify how we do it when given complete access to $\boldsymbol{A}$ and linear space with building-$\boldsymbol{A}_*$ (Algorithm 1). Then we will show how to turn it into a linear sketch, accommodating insertions and deletions to the entries of $\boldsymbol{A}$.

We now prove that building-$\boldsymbol{A}_*$ correctly builds $\boldsymbol{A}_*$ with high probability. At a high level, we subsample down to a smaller universe $\boldsymbol{A}'$ which only causes us to lose an $\varepsilon$ factor in our approximation.

---

**Algorithm 1** building-$\boldsymbol{A}_*$ ($n \times d$ matrix $\boldsymbol{A}$, $\epsilon \in (0,1)$, $k$)

---

1: Set $\delta = (2 + \log d) \log \log_{1-\varepsilon} n$.
2: Set $\varepsilon = \varepsilon/8$.
3: Subsample rows from $\boldsymbol{A}$ to get $\boldsymbol{A}'$ such that **OPT** in $\boldsymbol{A}'$ is $O(k \log d/\varepsilon^2)$. For clarity, row $j$ in $\boldsymbol{A}'$ and $\boldsymbol{A}$ both correspond to the row vector that corresponds to item $j$.
4: Set $b = O(\frac{k \log d}{\varepsilon^2})$.
5: Set $t = O(\log d)$.
6: **for** $i = 1, \ldots, t$ **do**
7:     Use a hash function to hash each subsampled row of $\boldsymbol{A}'$ to $b$ buckets in structure $\mathcal{C}_i$.
8:     **for** each bucket in $\mathcal{C}_i$ **do**
9:         If there are $r$ rows hashed to the bucket, denote the $r$ rows concatenated into a vector of length $rd$ as $\boldsymbol{v}$.
10:         Randomly sample $O(\frac{d \log(1/\varepsilon)}{\varepsilon k})$ nonzero entries from $\boldsymbol{v}$ and store it in $\boldsymbol{A}'_i$.
11:     **end for**
12: **end for**
13: Initialize $\boldsymbol{A}_*(k, \varepsilon)$ as a $n \times d$ matrix with all entries initially set to 0.
14: Let $\mathcal{P}$ be a random permutation of the rows that are in $\boldsymbol{A}'$.
15: **while** the number of nonzero entries in $\boldsymbol{A}_*(k, \varepsilon)$ is less than $\frac{24 d \delta' \log(1/\varepsilon) \log d}{(1-\varepsilon)\varepsilon^3}$ **do**
16:     Process the row $j$ that comes next in $\mathcal{P}$.
17:     Determine among all $i \in [t]$ which $\boldsymbol{A}'_i$ has the most nonzero entries from row $j$. Take this $i$ to be $z$.
18:     **if** row $j$ has less than $\frac{d \log(1/\varepsilon)}{\varepsilon k}$ nonzero entries in $\boldsymbol{A}'_z$ **then**
19:         Add all of the nonzero entries from row $j$ in $\boldsymbol{A}'_z$ to $\boldsymbol{A}_*(k, \varepsilon)$.
20:     **else**
21:         Add $\frac{d \log(1/\varepsilon)}{\varepsilon k}$ of the nonzero entries from row $j$ in $\boldsymbol{A}'_z$, chosen arbitrarily, to $\boldsymbol{A}_*(k, \varepsilon)$.
22:     **end if**
23: **end while**

---

Now in this smaller universe, we hash the rows to a bunch of buckets. In each bucket, we will keep a number of nonzero entries and add them to $\boldsymbol{A}_*$. We do the process of hashing the rows to buckets for $t$ iterations. We will prove that these rows are sufficiently spread out ensuring that no bucket contains too many rows with nonzero entries. This means that for each row in $\boldsymbol{A}'$ that has nonzero entries, in one of the $i \in [t]$ iterations, $\boldsymbol{A}'_i$ will hold $\tilde{O}(d/k)$ of its nonzero entries.

**Claim 3.1.** *Obtaining an $(1 - 1/e)$ approximate solution to maximum coverage on $\boldsymbol{A}'$ is an $(1 - 1/e - \varepsilon/4)$ approximation solution on $\boldsymbol{A}$ with probability at least $1 - 1/\text{poly}(d)$.*

We denote items (or rows) of $\boldsymbol{A}'$ that have at least $d/k$ nonzero entries as "large" and the others as "small". We argue that the number of large items and the total number of nonzero entries among small items is bounded appropriately.

**Lemma 3.2.** *There are at most $O(k \log d/\varepsilon^2)$ large items in $\boldsymbol{A}'$.*

**Lemma 3.3.** *There are $O(\frac{d \log d}{\varepsilon^2})$ total nonzero entries among small items in $\boldsymbol{A}'$.*

We want to show that for each large item, we recover $d \log(1/\varepsilon)/(\varepsilon k)$ of their nonzero entries from $\boldsymbol{A}'$. In addition, we want to show that for each small item, we recover all their nonzero entries from $\boldsymbol{A}'$. We refer to any item corresponding to a row in $\boldsymbol{A}'$ that contains nonzero entries as a "nonzero" item. We begin by proving that each nonzero item is hashed to a bucket with no other large item with high probability.

**Claim 3.4.** *Every nonzero item for some $i \in [t]$ is hashed to a bucket with no other large item with probability $1 - 1/\text{poly}(d)$.*

We also want each nonzero item to be hashed to a bucket that does not have too many nonzero entries from small items.

**Claim 3.5.** *Every nonzero item for some $i \in [t]$ is hashed to a bucket containing at most $O(\frac{d \log(1/\varepsilon)}{\varepsilon k})$ nonzero entries from small items with probability $1 - 1/\text{poly}(d)$.*

**Lemma 3.6.** *We recover all nonzero entries from small items and $d \log(1/\varepsilon)/(\varepsilon k)$ nonzero entries from each large item present in $\boldsymbol{A}'$ with probability $1 - 1/\mathrm{poly}(d)$.*

We now show how to implement building-$\boldsymbol{A}_*$ via a linear sketch, Max-Coverage-LS (Algorithm 5). Again recall that we must build $\boldsymbol{A}_*$ while receiving updates to the entries of underlying matrix $\boldsymbol{A}$. Next we run max-coverage (Algorithm 6) on $\boldsymbol{A}_*$, setting the appropriate parameters and apply the greedy algorithm to obtain the final solution. We defer these algorithms to Appendix A.2.

**Lemma 3.7.** *Max-Coverage-LS (Algorithm 5) and max-coverage (Algorithm 6) correctly implement building-$\boldsymbol{A}_*$(Algorithm 1) and $k$-cover (Algorithm 4) with probability at least $1 - 1/\mathrm{poly}(d)$.*

**Claim 3.8.** *Maximum-Coverage-LS can be implemented using $\tilde{O}(d/\varepsilon^3)$ bits of memory.*

**Claim 3.9.** *The update time of Maximum-Coverage-LS is $\tilde{O}(d/\varepsilon^3)$ and the total reporting time (including running max-coverage) is $\tilde{O}(kd/\varepsilon^3)$.*

With Lemma 3.7, Claim 3.8, and Claim 3.9, we can now conclude the proof. Note that we incur only a $\varepsilon$ factor loss in total, resulting in a final $1 - 1/e - \varepsilon$ approximation. Specifically, we lose a $\varepsilon/4$ factor going from $\boldsymbol{A}$ to $\boldsymbol{A}'$, another $\varepsilon/4$ factor from running the greedy algorithm on $\boldsymbol{A}_*$, and a $\varepsilon/4$ factor from using the $L_0$ sketches to determine which set of outputs to return. Our sketch is directly applicable to turnstile streams. We can run the sketch during the stream, handling all insertions and deletions as they occur. Once the stream is complete, running max-coverage (Algorithm 6) will give Corollary 1.1.

## 4  A LINEAR SKETCH FOR $n^p - F_p$ FOR INTEGERS $p \geq 2$

We now prove Theorem 3 with p-Tuples-Sketch (Algorithm 2). Recall that we are given a $n$-dimensional vector $\mathbf{x}$ where we denote $\mathcal{Z}$ as the set of distinct values in vector $\mathbf{x}$ and $f_i$ is the frequency of the $i^{\text{th}}$ distinct value in $\mathbf{x}$. For example, take $\mathbf{x} = (1, 5, 5, 3, -2, 3, 3, 7, 3)$. Here the distinct values are $1, 5, 3, -2$, and $7$ and the respective frequencies of those values are $1, 2, 4, 1$, and $1$. Our goal is to compute $n^p - \sum_{i \in \mathcal{Z}} f_i^p$. Updates are of the form $(i, \pm 1)$ which modifies $\mathbf{x}_i$ by adding or subtracting $1$. We now present our algorithm. At a high level, we keep $L_0$ sketches and perfect $L_0$ samplers. If there is a value with frequency $\Theta(n)$, we use a $L_0$ sketch to estimate its frequency. Otherwise, we use the $L_0$ samplers, which provide uniform samples of the nonzero entries of a vector, to estimate the frequencies of the rest of the values. For values with very small frequency, we ignore them and show this does not result in too much error. We defer the proof to Appendix A.5.

## 5  EXPERIMENTS

We first outline our fingerprinting results and compare the runtime/accuracy to Gulyás et al. (2016) [3]. We then present our results on dimensionality reduction. All experiments were run locally on a M2 MacBook Air, with code shared on Google Colab for distribution. We use two publicly-available datasets, the UC Irvine "Adult" and "US Census Data (1990)" Becker & Kohavi (1996); Meek et al.. For consistency, we apply the pre-processing from Gulyás et al. (2016) to both datasets. The pre-processed dataset of "Adult" has $32,561$ instances (representing users) and $80$ features. While the original dataset has $15$ features, Gulyás et al. (2016) empirically treats each value of each attribute as a separate attribute. So instead of the attribute being "workclass", each potential value of "workclass" is its own attribute. The second dataset we use, "US Census Data (1990)", has $2,458,285$ instances and $68$ original features. We treat attributes the same as above. Therefore, our input matrix $\boldsymbol{A}$ is an $n = 2,458,285$ by $d = 195$ matrix.

**Targeted Fingerprinting Results.** We note the differences between our theoretical and implemented algorithm. We make standard modifications done in the practical implementation of streaming algorithms. In particular, we use a constant subsampling rate $p \in [0.1, 0.6]$ instead of subsampling at $\log n$ rates, and we sample nonzero entries once we are in the smaller subsampled universe with a fixed probability as this is sufficient for smaller datasets. We first present our results for the UCI "Adult" dataset. We present results for subsampling rows from $\boldsymbol{A}$ to create $\boldsymbol{A}'$ with $p = 0.1, 0.2, 0.4$, and $0.6$. One run corresponds to finding the targeted fingerprint of all users in the

---

[3]Gulyás et al. (2016) has two implementations, one of which is supposed to be optimized for time. However, we found that the non-optimized implementation was faster and therefore use it for comparison.

---

**Algorithm 2** p-Tuples-Sketch ($n \times 1$ vector $\mathbf{x}$, constant integer $p \geq 2$, $\gamma, \delta \in (0,1)$)

1: $\varepsilon \leftarrow \frac{\gamma^{\frac{1}{p-1}}}{16 \cdot 2^p}$.
2: Keep three independent $L_0$ sketches, $L_0^1, L_0^2, L_0^3$ of $\mathbf{x}$ each with $\delta' = \delta/8$ and $\varepsilon = \varepsilon$.
3: Keep $t = 2/\varepsilon^2 \cdot \log(2(\delta/8)^{-1})$ perfect $L_0$ samplers of $\mathbf{x}$ and concatenate them into $\mathcal{S}$.
4: Set $\delta'$ for each $L_0$ sampler s.t. the total probability of failure across them is at most $\delta/8$.
5: Upon an update, the $L_0$ sketches and perfect $L_0$ samplers will handle updates.
6: **Upon a query:**
7: Initialize an empty set $\mathcal{B}$.
8: Query $L_0^1$ sketch to get $w_1$ and set $b = 0$ and $f_b' = n - w_1$.
9: Query $L_0^2$ to get $w_2$. Estimate the frequency of a value using $\mathcal{S}$ by taking its frequency in $\mathcal{S}$ and scaling by $w_2/t$.
10: Find the value $v$ with highest frequency $f'$ in $\mathcal{S}$.
11: **if** $f' > f_b'$ **then**
12:     Set $b = v$ and $f_b' = f'$.
13: **end if**
14: **if** $f'b < \frac{3\gamma^{\frac{1}{p-1}}}{4} \cdot n$ **then**
15:     Output $n^{\frac{1}{p}}$.
16: **end if**
17: **if** $f'b > \frac{n}{2}$ **then**
18:     Subtract off value $b$ from $L_0^3$ and query it to get $w_3$.
19:     Set $f_b' = n - w_3$.
20: **end if**
21: Add $(b, f_b')$ to $\mathcal{B}$.
22: Take $t$ perfect $L_0$ samplers of a $n$-dimensional vector with each entry set to value $b$ and concatenate them to form $\mathcal{S}_b$.
23: Set $\delta'$ for each $L_0$ sampler in $\mathcal{S}_b$ s.t. the total probability of failure across them is at most $\delta/8$.
24: $\mathcal{S}_* \leftarrow \mathcal{S} - \mathcal{S}_b$.
25: Use $\mathcal{S}_*$ to get all values and their frequencies (take the frequency in $\mathcal{S}_*$ and scale by $f_b'/t$).
26: **for** all values $v$ with frequency $f_v' \geq \frac{\gamma^{\frac{1}{p-1}}}{4}(n - f_b')$ **do**
27:     Add $(v, f_v')$ to $\mathcal{B}$.
28: **end for**
29: Using all $z'$ tuples $(v, f_v') \in \mathcal{B}$, calculate $n^p - \sum_{j=1}^{z'} (f_j')^p$ and output.

---

dataset for some given cardinality constraint $k$. First we look at the running time of our algorithm compared to Gulyás et al. (2016). We have $k = 7$ here. The following are averages over 10 runs.

From fig. 1, our algorithm runs about 25x, 8.4x, 3x, and 2.3x faster than that of Gulyás et al. (2016) with subsampling probabilities 0.1, 0.2, 0.4, and 0.6 respectively. In settings where $n$ is very large the subsampling probability in our algorithm will be much smaller. We only run our algorithm with larger subsampling probabilities for further insight. Note that the implementation of Gulyás et al. (2016) is deterministic. We put their runtime as a line for visualization. Now we look at accuracy. For increasing $k$, we compute the average percent of users our algorithm is able to separate from a given target user and compare it to the algorithm of Gulyás et al. (2016). In fig. 1,

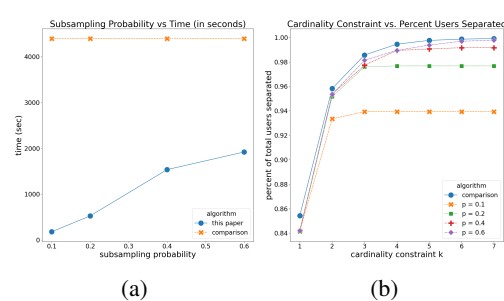

(a)         (b)

Figure 1: Comparison: Gulyás et al. (2016).

we show that we retain good accuracy despite subsampling rows and then subsampling nonzero entries. Note that the vertical axis's minimum value is $84\%$. As the subsampling probability increases, the accuracy of our implementation converges to that of Gulyás et al. (2016). We again note that we took an average over 10 runs.

Now, we present our results for the UCI "US Census Data (1990)" dataset. Due to limited compute, we look at one subsampling level of $0.1$. For comparing the time of our algorithm and the previous work of Gulyás et al. (2016), we again use $k = 7$. Over 10 runs, the average time of our implementation to compute a fingerprint for an input user is $1.06$ seconds while the comparison average time is $52.6$ seconds. The subsampling took an extra $46.355$ seconds. This means that our implementation is about 49x times faster. We measure accuracy the same way as for the previous dataset. We can see in fig. 2 that we quickly converge to the accuracy of Gulyás et al. (2016) with growing $k$. Note that the vertical axis's minimum value is $92\%$.

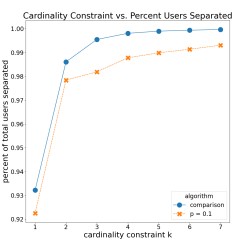

Figure 2

**General Fingerprinting Results.** The main difference between our theoretical and implemented algorithm is that we only create one sketch rather than $k$ sketches. We first present our results for the "Adult" dataset. The main variable we vary in our experiments is the size of our $L_0$ sketches. We present results for an $L_0$ sketch with $300, 600, 900$, and $1,250$ rows. We had our algorithm compute a general fingerprint for $k = 1, 2, \ldots, 20$ to compare with Gulyás et al. (2016). The runtime of our algorithm slightly increased as the sketch size increased. However for all sketch sizes it ran in about $0.8$ seconds which is 44x faster than the $35.30$ second runtime of Gulyás et al. (2016). Now we consider the accuracy of our algorithm. We measure accuracy by looking at the proportion between the number of pairs of users that our algorithm separates to the number of pairs of users that the algorithm from Gulyás et al. (2016) separates. For each sketch size, we never dip below an accuracy ratio of $80\%$, and as the sketch size increases the accuracy ratio increases to around $99\%$. We now present our results for the "US Census Data (1990)" dataset. We vary the size of our $L_0$ sketches, this time with $55,000$ rows, $180,000$ rows, and $400,000$ rows. We computed a general fingerprint for $k = 1, 2, \ldots, 10$. We use smaller $k$ for comparison for this dataset since the implementation of Gulyás et al. (2016) was not able to terminate even after several hours for larger $k$.

These are averages over 10 runs. In fig. 3, the runtime of our algorithm increases as the sketch size increases. Our implementation is about 210, 120, and 45 times faster than that of Gulyás et al. (2016) for $55,000$, $180,000$, and $400,000$ rows respectively. For a fingerprint of size 20 our implementation takes a little over twice the amount of time as for a fingerprint of size 10 shown here. We estimate that the runtime of the comparison algorithm also doubles but cannot be sure due to its non-termination. We measure accuracy in the same way as the previous dataset. We again see in fig. 3 that as sketch size increases, the accuracy ratio increases. We make note of a steep drop-off for a sketch with $55,000$ rows. However, our accuracy ratio never dips below $70\%$.

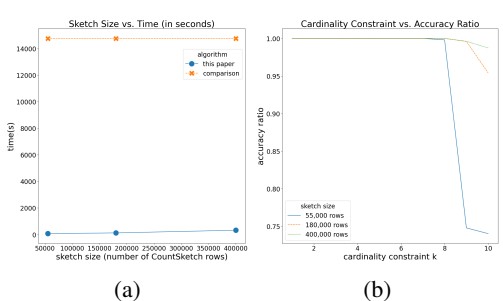

(a)          (b)

Figure 3: Comparison: Gulyás et al. (2016).

**Dimensionality Reduction Results.** We use the UCI "Wine" dataset which consists of 178 instances and 13 features Aeberhard & Forina (1991). Each of the instances is labeled by one of three wine types. We used our general fingerprinting algorithm to select features that best separate the data. Then, we ran $k$-means with 3 clusters (for the 3 wine types) using just the selected features. Therefore, this is a dimensionality reduction technique, since for many clustering algorithms (including $k$-means and $k$-means++) the efficiency depends on the feature dimension. We measure accuracy in the following way. After running $k$-means on the reduced feature space, for each cluster, we calculate the majority wine type. Then, for each instance, if its actual wine type is not the same as the majority wine type of its assigned cluster, we count it towards the error. We used general fingerprinting to reduce the feature dimension to $3, 4$, and $5$ features. Our accuracy for all was around $68\%$. When running $k$-means using all 12 features, the accuracy was around $71\%$, which suggests that we do not introduce that much error. In addition, when running $k$-means instead on just $3, 4$, and $5$ completely randomly chosen features, the accuracy decreases to around $52\%$. We also increase the efficiency of running $k$-means. Running $k$-means with our reduced $3, 4$, and $5$ features compared to running it with all 13 features is about $3.2, 2.4$, and $2.1$ times faster, respectively.

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

# A  APPENDIX

## A.1  EXTENDED PRELIMINARIES

### A.1.1  TURNSTILE STREAMING MODEL

In this paper, we represent the input as a $n \times d$ matrix $\boldsymbol{A}$. In the streaming model, it is standard to initialize all the entries to zero before the stream of updates. The algorithm then processes a stream of updates which come one-by-one, each of the form $(i, j, \pm 1)$. This modifies entry $\boldsymbol{A}_{ij}$ by performing $\boldsymbol{A}_{ij} = \boldsymbol{A}_{ij} + 1$ or $\boldsymbol{A}_{ij} = \boldsymbol{A}_{ij} - 1$ depending on the sign. This is referred to as the turnstile streaming model, where both insertions and deletions (or positive and negative updates) are allowed. The updates can appear in arbitrary order in the stream, and we make the standard assumption that the length of the stream is at most $\text{poly}(n)$. The goal of the streaming algorithm is to process the stream efficiently, using sublinear space in the size of the input matrix $\boldsymbol{A}$ (and therefore cannot store all the updates) and a small constant number of passes over the stream. In this work, restrict our focus to one-pass algorithms. At the end of the stream, the algorithm can do some post-processing and then must output the answer. While streaming algorithms are not required to maintain a stored answer at every point during the stream, there is no restriction on when the stream may terminate. Any time or space used before or after processing the stream is attributed to pre-processing or post-processing, respectively. Generally, our primary focus is on optimizing the memory usage and update time during the stream. Here the update time is the time complexity required by the algorithm to process an update.

### A.1.2  USEFUL DEFINITIONS

$\ell_0$ **Norm.**  Consider an underlying vector $\mathbf{x} = (\mathbf{x}_1, \mathbf{x}_2, \ldots, \mathbf{x}_n)$. The $\ell_0$ norm of $\mathbf{x}$ is the number of non-zero entries in $\mathbf{x}$. Formally, it is $\|\mathbf{x}\|_0 = \sum_{i=1}^{n} \mathbb{I}(\mathbf{x}_i \neq 0)$. The $\ell_0$ norm is not a proper norm since it does not meet the homogeneity requirement. However, it is still a well-defined quantity.

**Submodular Maximization.**  Consider a non-negative set function $f : 2^V \to \mathbb{R}_+$. If for all $S \subseteq T \subseteq V \setminus \{e\}$, $f$ satisfies: $f(S \cup \{e\}) - f(S) \geq f(T \cup \{e\}) - f(T)$, then $f$ is submodular. We assume that $f(\emptyset) = 0$. If $f(S) \leq f(T)$ for all $S \subseteq T$, then $f$ is also monotone. When $f$ is submodular and monotone, we aim to solve $\max_{|\mathcal{C}| \leq k} f(\mathcal{C})$ given a cardinality constraint $k$.

**Linearly Sketchable Functions.** All the functions $f : 2^d \to \mathbb{R}_+$ that we consider are of the form $f(\mathcal{C}) = g(\{a_i\}_{i \in \mathcal{C}})$ where $a_1, \ldots, a_d$ are a set of vectors that are either fixed in advance or are the columns of the $n \times d$ matrix $\boldsymbol{A}$ that are being updated in the stream. We say that a function $f$ is "linearly sketcheable" if there exists a randomized sketching matrix $\boldsymbol{S}$ and a corresponding function $g_{\boldsymbol{S}}$ such that, for any vectors $a_1, \ldots, a_d$, with high probability for all $\mathcal{C} \subseteq [d]$, $f(\mathcal{C})$ can be approximated by $g_{\boldsymbol{S}}(\{\boldsymbol{S} \cdot a_i\}_{i \in \mathcal{C}})$.

### A.1.3 CONCENTRATION INEQUALITIES

**Markov's Inequality.** If $X$ is a nonnegative random variable and $a > 0$, then

$$\Pr(X \geq a) \leq \frac{\mathbb{E}[X]}{a}.$$

**Chebyshev's Inequality.** For any random variable $X$ and $t > 0$.

$$\Pr(|X - \mathbb{E}[X]| \geq t) \leq \frac{\mathrm{Var}[X]}{t^2}.$$

### A.2 DEFERRED PARTS OF SECTION 3 (MAXIMUM COVERAGE)

We restate the algorithm of Bateni et al. (2017), $\boldsymbol{A}_*(k, \varepsilon, \delta)$ (Algorithm 3).

---

**Algorithm 3** $\boldsymbol{A}_*(k, \varepsilon, \delta)$

---

**Require:** $k, \varepsilon \in (0, 1]$, and $\delta$.
**Ensure:** $\boldsymbol{A}_*(k, \varepsilon, \delta)$.
 1: Let $\delta' = \delta \log \log_{1-\varepsilon} n$.
 2: Let $h$ be an arbitrary hash function that uniformly and independently maps each item (or each row of $\boldsymbol{A}$) to $[0, 1]$.
 3: Initialize $\boldsymbol{A}_*(k, \varepsilon, \delta)$.
 4: **while** number of nonzero entries in $\boldsymbol{A}_*(k, \varepsilon, \delta)$ is less than $\frac{24 d \delta' \log(1/\varepsilon) \log d}{(1-\varepsilon)\varepsilon^3}$ **do**
 5:     Pick item $i$ of minimum $h(i)$ that has not been considered yet.
 6:     **if** there are less than $\frac{d \log(1/\varepsilon)}{\varepsilon k}$ nonzero entries in the $i^{\text{th}}$ row of $\boldsymbol{A}$ **then**
 7:         Add all the nonzero entries from the $i^{\text{th}}$ row of $\boldsymbol{A}$ to $\boldsymbol{A}_*(k, \varepsilon, \delta)$.
 8:     **else**
 9:         Add $\frac{d \log(1/\varepsilon)}{\varepsilon k}$ of the nonzero entries of $\boldsymbol{A}$, chosen arbitrarily, to $\boldsymbol{A}_*(k, \varepsilon, \delta)$.
10:     **end if**
11: **end while**

---

We now restate the final algorithm from Bateni et al. (2017), $k$-cover(Algorithm 4).

---

**Algorithm 4** $k$-cover

---

**Require:** $k$ and $\varepsilon \in [0, 1]$.
**Ensure:** A $(1 - 1/e - \varepsilon)$ approximate solution to maximum coverage with probability $1 - 1/d$.
 1: Set $\delta = 2 + \log d$ and $\varepsilon' = \varepsilon/12$.
 2: Construct sketch $\boldsymbol{A}_*(k, \varepsilon', \delta)$.
 3: Run the greedy algorithm (or any $1 - 1/e$ approximation algorithm) on $\boldsymbol{A}_*(k, \varepsilon', \delta)$ and report the output.

---

**Claim 3.1.** *Obtaining an $(1 - 1/e)$ approximate solution to maximum coverage on $\boldsymbol{A}'$ is an $(1 - 1/e - \varepsilon/4)$ approximation solution on $\boldsymbol{A}$ with probability at least $1 - 1/\mathrm{poly}(d)$.*

*Proof.* This states that we only lose a $\varepsilon/4$ factor by reducing to a smaller universe via subsampling such that $\mathbf{OPT} = O(k \log d/\varepsilon^2)$. This is proven in McGregor & Vu (2018) in Corollary 9. Note that in McGregor & Vu (2018) they prove a $(1 - 1/e)$ approximate solution on $\boldsymbol{A}'$ is an $(1 - 1/e - 2\varepsilon)$-approximation solution on $\boldsymbol{A}$ but we re-weigh $\varepsilon$ in our algorithm. $\qquad\square$

**Lemma 3.2.** *There are at most $O(k \log d/\varepsilon^2)$ large items in $\boldsymbol{A'}$.*

*Proof.* Suppose that there are $\ell$ large items. First, we choose a subset (of the $d$ input subsets) which covers $c_1$ of those $\ell$ large items. We then remove that subset and the $c_1$ large items it covered. We continue the process by choosing a subset which covers $c_2$ large items and so on for a total of $k$ times.

In total, we know that $c_1 + c_2 + \cdots + c_k = C_1 \cdot k \log d/\varepsilon^2$ since we have that $\mathbf{OPT} = C_1 \cdot k \log d/\varepsilon^2$ for some constant $C_1$. Now, suppose for the sake of contradiction that $\ell = C_2 \cdot k \log d/\varepsilon^2$ for some constant $C_2$. Then,

$$C_2 \cdot k \log d/\varepsilon^2 - c_1 - \cdots - c_k > C_2 \cdot k \log d/\varepsilon^2 - C_1 \cdot k \log d/\varepsilon^2 > C_2 \cdot k \log d/(2\varepsilon^2)$$

for $C_2 > 2C_1$.

So, at each step in the above process, there are at least $C_2 \cdot k \log d/(2\varepsilon^2)$ large items, and hence, at least $C_2 d \log d/(2\varepsilon^2)$ nonzero entries among large items. So in each step, we should have been able to find a subset covering at least $C_2 \log d/(2\varepsilon^2)$ additional large items. This means at the end of the process (choosing $k$ times a subset which covers some number of large items and then removing the items and subset) we will have covered at least $C_2 k \log d/(2\varepsilon^2)$ items. But

$$\mathbf{OPT} = C_1 \cdot k \log d/\varepsilon^2 < C_2/2 \cdot k \log d/\varepsilon^2$$

so we have a contradiction. $\qquad\square$

**Lemma 3.3.** *There are $O(\frac{d \log d}{\varepsilon^2})$ total nonzero entries among small items in $\boldsymbol{A'}$.*

*Proof.* Suppose that there are $s$ total nonzero entries among small items. We first find a subset (out of the $d$ input subsets) that covers $c_1$ small items. Then we remove that subset along with the $c_1$ small items. Note that we remove at most $c_1 \cdot d/k$ nonzero entries. We can then find a subset that covers $c_2$ small items and remove that subset and those $c_2$ small items. We keep on doing this for $k$ subsets total.

Suppose for the sake of contradiction that $s = C_1 \cdot d \log d/\varepsilon^2$ for some constant $C_1$. So, in each step in the above process, we could have removed at least $C_1 \log d/\varepsilon^2$ nonzero entries. However this means that $\mathbf{OPT} \geq C_1 k \log d/\varepsilon^2$, which for appropriate $C_1$ contradicts $\mathbf{OPT} = O(k \log d/\varepsilon^2)$. $\qquad\square$

**Claim 3.4.** *Every nonzero item for some $i \in [t]$ is hashed to a bucket with no other large item with probability $1 - 1/\mathrm{poly}(d)$.*

*Proof.* Take some nonzero item $x$. By Lemma 3.2, there are at most $C_1 \cdot k \log d/\varepsilon^2$ large items for some constant $C_1$. For each $i \in [t]$, $\mathcal{C}_i$ has $C_2 \cdot k \log d/\varepsilon^2$ buckets. For appropriate $C_2$, we can say that $C_2 > 2C_1$. In the worst case, every large item (besides potentially large item $x$) is hashed to a different bucket. Then for each $i$ the probability of $x$ being hashed to a bucket with another large item is at most $1/2$. Note that we hash $O(\log(d))$ times (since we do it for $i \in [t]$). Since we have at most $\tilde{O}(k + d)$ nonzero items by Lemma 3.2 and Lemma 3.3, we have the result by taking a union bound. $\qquad\square$

**Claim 3.5.** *Every nonzero item for some $i \in [t]$ is hashed to a bucket containing at most $O(\frac{d \log(1/\varepsilon)}{\varepsilon k})$ nonzero entries from small items with probability $1 - 1/\mathrm{poly}(d)$.*

*Proof.* Take some nonzero item $x$. It suffices to show with high probability that not too many (nonzero) small items are hashed to the same bucket as $x$ for every $i \in [t]$. For some $i$, take the bucket that $x$ was hashed to as $b_i$. The expected number of nonzero entries in $b_i$ from small items is at most $C \cdot d/k$ for some constant $C$ since by Lemma 3.3 there are at most $O(d \log d/\varepsilon^2)$ total nonzero entries among small items, and we hash to $O(k \log d/\varepsilon^2)$ buckets.

By Markov's inequality, the probability that the true number of nonzero entries in $b_i$ from small items is more than $2C \cdot d/k$ is at most $1/2$. Note that we have $O(d \log(1/\varepsilon)/(\varepsilon k)) \geq 2C \cdot d/k$ for $\varepsilon \in (0, 1/2)$. However note that this still extends for the full range of $\varepsilon$ since we can always use

a smaller $\varepsilon$ to achieve the desired error bound while only incurring an extra constant factor in the space/time.

Since we hash $O(\log d)$ times (for $i \in [t]$), taking a union bound over the total number of nonzero items in $\boldsymbol{A}'$ gives us the result. $\qquad\square$

**Lemma 3.6.** *We recover all nonzero entries from small items and $d\log(1/\varepsilon)/(\varepsilon k)$ nonzero entries from each large item present in $\boldsymbol{A}'$ with probability $1 - 1/\mathrm{poly}(d)$.*

*Proof.* The fact that we recover all the nonzero entries of small items with probability $1 - 1/\mathrm{poly}(d)$ follows from Claim 3.5. The fact that we recover $d\log(1/\varepsilon)/(\varepsilon k)$ nonzero entries from each large item with probability $1 - 1/\mathrm{poly}(d)$ follows from Claim 3.4 and Claim 3.5. This is because each large item for some $i \in [t]$ is not hashed with another large item, and the number of nonzero entries from small items is at most $O(d\log(1/\varepsilon)/(\varepsilon k))$. Note that we have a constant number of events that each happen with probability $1 - 1/\mathrm{poly}(d)$. Taking a union bound over these events, we achieve overall probability of success at least $1 - 1/\mathrm{poly}(d)$. $\qquad\square$

We show how to implement building-$\boldsymbol{A}_*$ via a linear sketch, Max-Coverage-LS (Algorithm 5).

Then we perform the following process, max-coverage (Algorithm 6), mostly revolving around running the greedy algorithm to get the final answer.

**Lemma 3.7.** *Max-Coverage-LS (Algorithm 5) and max-coverage (Algorithm 6) correctly implement building-$\boldsymbol{A}_*$(Algorithm 1) and $k$-cover (Algorithm 4) with probability at least $1 - 1/\mathrm{poly}(d)$.*

*Proof.* The first step in building-$\boldsymbol{A}_*$ is subsampling from $\boldsymbol{A}$ to get $\boldsymbol{A}'$ such that $\mathbf{OPT}$ in $\boldsymbol{A}'$ is $O(k\log d/\varepsilon^2)$. Since this sampling rate depends on what $\mathbf{OPT}$ is in $\boldsymbol{A}$, in Max-Coverage-LS, we instead sample in $\log n$ different rates. So in one of the $\log n$ different parallel runs, we will sample with the correct rate. We will describe how we choose the right run to consider later.

Let us consider the parallel run with the correct sampling rate. The rest of Max-Coverage-LS is identical to building-$\boldsymbol{A}_*$. The only difference is that in Max-Coverage-LS we are uniformly sampling nonzero entries using perfect $L_0$ samplers. The correctness follows from the correctness of the perfect $L_0$ samplers. We set the failure probability appropriately for the $L_0$ samplers and $L_0$ sketches so we only incur $1/\mathrm{poly}(d)$ total error.

So Max-Coverage-LS (Algorithm 5) produces a $L_0$ sketch for each column of $\boldsymbol{A}$ and $\boldsymbol{A}_{m,*}$ for $m \in [\log n]$. We must figure out which $\boldsymbol{A}_{m,*}$ is the one that corresponds to the desired subsampling rate. We instead find which $\boldsymbol{A}_{m,*}$ gives us the best answer on the original input $\boldsymbol{A}$ using the $L_0$ sketches in the following way.

Suppose that for some $\boldsymbol{A}_{m,*}$ the greedy algorithm chooses subsets $s_1, \ldots, s_{,}k$. We take the $L_0$ sketches for these subsets (or columns of $\boldsymbol{A}$) and reduce to the vector case to estimate how many distinct items these subsets cover in their union.

Imagine that we are working with the original input $\boldsymbol{A}$. Now, take the original columns $s_1, \ldots, s_k$ and concatenate them into a $n \times k$ matrix $\boldsymbol{L}$. We now randomly generate a $k \times 1$ vector $\mathbf{x}$ with entries between $[-\mathrm{poly}(d), \mathrm{poly}(d)]$. Now multiply $\boldsymbol{L}$ by $\mathbf{x}$. We can see with probability at least $1 - 1/\mathrm{poly}(d)$, the $i^{\text{th}}$ entry in $\boldsymbol{L} \cdot \mathbf{x}$ is nonzero if and only if the $i^{\text{th}}$ row of $\boldsymbol{L}$ is nonzero. So, if the $i^{\text{th}}$ entry of $\boldsymbol{L} \cdot \mathbf{x}$ is nonzero, that means the $i^{\text{th}}$ item was covered by the union of subsets $s_1, \ldots, s_k$.

Note that $\boldsymbol{L} \cdot \mathbf{x}$ is by definition equivalent to summing $\boldsymbol{L}_1 \cdot \mathbf{x}_1 + \boldsymbol{L}_2 \cdot \mathbf{x}_2 + \cdots + \boldsymbol{L}_k \cdot \mathbf{x}_k$ where $\boldsymbol{L}_i$ denotes the $i^{\text{th}}$ column of $\boldsymbol{L}$ and $\mathbf{x}_i$ denotes the $i^{\text{th}}$ entry of $\mathbf{x}$. Since the $L_0$ sketches are linear sketches, by definition they have the property the $L_0$ sketch of the sum of two vectors is equivalent to summing the $L_0$ sketches for the two vectors [4]. Therefore, using the $L_0$ sketches we can create the $L_0$ sketch for $\boldsymbol{L} \cdot \mathbf{x}$ and query it to get a $(1 + \varepsilon/4)$ approximation to the true coverage of the union of subsets $s_1, \ldots, s_k$. $\qquad\square$

**Claim 3.8.** *Maximum-Coverage-LS can be implemented using $\tilde{O}(d/\varepsilon^3)$ bits of memory.*

---

[4]See Section 2.

---

**Algorithm 5** Max-Coverage-LS ($n \times d$ matrix $\boldsymbol{A}$, $\epsilon \in (0,1)$, $k$)

---

1: Set $\delta = (2 + \log d) \log \log_{1-\varepsilon} n$.
2: Set $\varepsilon = \varepsilon/8$.
3: Keep a $L_0$ sketch for each column of $\boldsymbol{A}$. Denote these as $L_0(j)$ for $j \in [d]$.
4: **for** $m = 1, 2, \ldots, \log n$ **do**
5:    {Run in parallel}
6:    Use a hash function to subsample rows from $\boldsymbol{A}$ with probability $1/2^m$. Call the subsampled matrix we consider in this iteration $\boldsymbol{A}'_m$.
7:    {We do not store $\boldsymbol{A}'_m$ explicitly. We are simply saying we only consider updates to $\boldsymbol{A}'_m$ in this iteration.}
8:    Set $b = O(\frac{k \log d}{\varepsilon^2})$.
9:    Set $t = O(\log d)$.
10:    **for** $i = 1, \ldots, t$ **do**
11:       Initialize an empty structure $\mathcal{S}_i$.
12:       Use a hash function to hash each row of $\boldsymbol{A}'_m$ to $b$ buckets in structure $\mathcal{C}_{m,i}$.
13:       {We do not store the rows of $\boldsymbol{A}'_m$ explicitly in structure $\mathcal{C}_{m,i}$. Rather, each bucket only considers updates to the rows that were hashed there.}
14:       **for** each bucket in $\mathcal{C}_{m,i}$ **do**
15:          If there are $r$ rows hashed to the bucket, denote the $r$ rows concatenated into a vector of length $rd$ as $\boldsymbol{v}$.
16:          Keep $O(\frac{d \log(1/\varepsilon)}{\varepsilon k})$ perfect $L_0$ samplers for $\boldsymbol{v}$. Add these samplers to structure $\mathcal{S}_i$.
17:       **end for**
18:    **end for**
19: **end for**
20: Set the error probability for each $L_0$ sketch and sampler such that the total error across all of them is at most $1/\text{poly}(d)$.
21: Upon an update, the $L_0$ sketches and $L_0$ perfect samplers will handle it.
22: **Upon a query:**
23: **for** each $m \in [\log n]$ **do**
24:    Initialize $\boldsymbol{A}_{m,*}(k, \varepsilon)$.
25:    Let $h$ be a hash function that maps uniformly between $[0,1]$ the rows of $\boldsymbol{A}'_m$ that have been sampled from with the perfect $L_0$ samplers and placed in $\mathcal{S}_i$ for some $i \in [t]$.
26:    **while** the number of nonzero entries in $\boldsymbol{A}_{m,*}(k, \varepsilon)$ is less than $\frac{24 d \delta' \log(1/\varepsilon) \log d}{(1-\varepsilon)\varepsilon^3}$ **do**
27:       Process the row $j$ that comes next in the ordering as determined by hash function $h$.
28:       Determine among all $i \in [t]$ which $\mathcal{S}_i$ has the most nonzero entries from row $j$. Take this $i$ to be $z$.
29:       **if** row $j$ has less than $\frac{d \log(1/\varepsilon)}{\varepsilon k}$ nonzero entries in $\mathcal{S}_z$ **then**
30:          Add all of the nonzero entries from row $j$ in $\mathcal{S}_z$ to $\boldsymbol{A}_{m,*}(k, \varepsilon)$.
31:       **else**
32:          Add $\frac{d \log(1/\varepsilon)}{\varepsilon k}$ of the nonzero entries from row $j$ in $\mathcal{S}_z$, chosen arbitrarily, to $\boldsymbol{A}_{m,*}(k, \varepsilon)$.
33:       **end if**
34:    **end while**
35: **end for**
36: Output the $L_0$ samplers and $\boldsymbol{A}_{m,*}(k, \varepsilon)$ for $m \in [\log n]$.

---

*Proof.* We first analyze the memory of our sketch. We subsample in $\log n$ levels and run $\log n$ instances in parallel. In each instance, we store $O(b \log(d))$ buckets for $b = k \log d/\varepsilon^2$ and a constant number of hash functions that use only $O(\log n)$ space each. In each bucket we store $O(\frac{d \log(1/\varepsilon)}{\varepsilon k})$ perfect $L_0$ samplers. Since perfect $L_0$ samplers take $\tilde{O}(\log^2 n)$ space, we have a total complexity of $\tilde{O}(d/\varepsilon^3)$. $\qquad\square$

**Claim 3.9.** *The update time of Maximum-Coverage-LS is $\tilde{O}(d/\varepsilon^3)$ and the total reporting time (including running max-coverage) is $\tilde{O}(kd/\varepsilon^3)$.*

---

**Algorithm 6** max-coverage

---

**Require:** $k$ and $\varepsilon \in [0, 1]$.
**Ensure:** A $1 - 1/e - \varepsilon$ approximate solution to maximum coverage with probability $1 - 1/d$.
1: Set $\varepsilon' = \varepsilon/48$.
2: For $m \in [\log n]$, construct $\boldsymbol{A}_{m,*}(k, \varepsilon')$ using Max-Coverage-LS (Algorithm 5). Also store the $L_0$ sketches of the columns of $\boldsymbol{A}$ outputted by Algorithm 5.
3: Run the greedy algorithm (or any $1 - 1/e$ approximation algorithm) on each $\boldsymbol{A}_{m,*}(k, \varepsilon')$.
4: Use the $L_0$ sketches to determine for which $\boldsymbol{A}_{m,*}$ the greedy algorithm gave the best answer and output it.

---

*Proof.* The update time of each perfect $L_0$ sampler is $\text{poly}(\max(\log n, \log d))$. Since we have $\tilde{O}(d/\varepsilon^3)$ perfect $L_0$ samplers. the total update time for them is $\tilde{O}(d/\varepsilon^3)$. The update time for each $L_0$ sketch is $O(1)$, and we have $d$ of them. This gives a total update time for the $L_0$ sketches of $O(d)$, and an overall update time for the entire sketch of $\tilde{O}(d/\varepsilon^3)$.

Running the greedy algorithm on the produced sketches in max-coverage dominates the reporting time. This takes time $\tilde{O}(kd/\varepsilon^3)$ since we have $k$ rounds in the greedy algorithm and a total of $\tilde{O}(d/\varepsilon^3)$ total nonzero entries in a sketch. $\qquad\square$

### A.3 TARGETED FINGERPRINTING

Recall that in targeted fingerprinting we have an $n \times d$ input matrix $\boldsymbol{A}$ where there are $n$ users and $d$ features and entry $\boldsymbol{A}_{ij}$ represents the value the $i^{\text{th}}$ user has for the $j^{\text{th}}$ feature. Given a target user $u$, we want to output at most $k$ features such that the number of other users who do not have identical values at all $k$ features to $u$ is maximized.

**Claim A.1.** *Take $\boldsymbol{A}'$ to be $\boldsymbol{A}$ with the updates $\boldsymbol{A}_{ij} = \boldsymbol{A}_{ij} - \boldsymbol{A}_{uj}$ applied for all $i \in [n], j \in [d]$. For some union of subsets $\mathcal{U}$, the number of items covered on $\boldsymbol{A}'$ is equivalent to the number of users separated from the target on $\boldsymbol{A}$.*

*Proof.* For all $i \in [n]$, for any $j \in [d]$ such that $\boldsymbol{A}_{ij} = \boldsymbol{A}_{uj}$, we have $\boldsymbol{A}'_{ij} = 0$. Additionally, for all $i \in [n]$, for any $j \in [d]$ such that $\boldsymbol{A}_{ij} \neq \boldsymbol{A}_{uj}$, we have that $\boldsymbol{A}'_{ij}$ is nonzero.

In other words, for all users, for any feature where they shared the same value with the queried user $u$, this entry is now 0. In addition, for any feature where they did not share the same value with the queried user, this entry is now nonzero. We can see that the maximum coverage problem on $\boldsymbol{A}'$ exactly corresponds to finding $k$ features which separates the most users from target user $u$ on $\boldsymbol{A}$. $\qquad\square$

Algorithmically, we simply store the row that corresponds to target user $u$ in $O(d)$ space. In addition, we can simulate forming $\boldsymbol{A}'$ from $\boldsymbol{A}$ by sending updates to the maximum coverage sketch for $\boldsymbol{A}$. Therefore, the approximation factor, space, update time, and reporting time all follow from Theorem 1 giving us Corollary 1.2. This linear sketch is then directly applicable to turnstile streams giving us Corollary 1.3.

### A.4 PROOF OF THEOREM 2 (SUBMODULAR MAXIMIZATION FRAMEWORK)

Here, we outline a framework to design algorithms to maximize monotone non-negative submodular functions that are linearly sketchable subject to a cardinality constraint. At a high level we will receive a linear sketch of the input matrix $\boldsymbol{A}$ such that querying the sketch will produce the function's output value on some union of subsets. We then adapt the classical greedy algorithm for maximizing a monotone submodular function to query the linear sketches instead of accessing the input matrix directly.

We note that setting $\gamma = \varepsilon/k$ for many linear sketches introduces $\text{poly}(k)$ factors in the final memory usage. However, setting $\gamma = \varepsilon/k$ is provably necessary when performing submodular maximization over queried function values that are preserved up to a $(1 \pm \gamma)$ factor to achieve a $1 - 1/e - \varepsilon$

approximation (see Theorem 5 of Horel & Singer (2016)). Note that this applies to all algorithms that perform submodular maximization that have this property.

We now prove Theorem 2. Theorem 2 allows us to create an algorithm to maximize a *specific* monotone non-negative submodular function subject to a cardinality constraint by simply sketching the input $\boldsymbol{A}$ via a linear sketch that satisfies the properties of the theorem.

Let $\mathcal{C}$ be a subset of the column vectors of $\boldsymbol{A}$. In the following, $\{\boldsymbol{S} \cdot a_i\}_{i \in \mathcal{C}}$ can be thought of as the sketch of $\boldsymbol{A}$ restricted to $\mathcal{C}$. As described in Appendix A.1.2, we say that our function $f$ has a corresponding sketching matrix $\boldsymbol{S}$ and corresponding $g_{\boldsymbol{S}}$. For any two subsets of columns $X$ and $Y$, let $g_{\boldsymbol{S}}(\{\boldsymbol{S} \cdot a_i\}_{i \in X|Y})$ denote the marginal gain of adding $X$, or $g_{\boldsymbol{S}}(\{\boldsymbol{S} \cdot a_i\}_{i \in X \cup Y}) - g_{\boldsymbol{S}}(\{\boldsymbol{S} \cdot a_i\}_{i \in Y})$. $c \in d \setminus \mathcal{C}$ denotes a column $c$ which is not already in subset $\mathcal{C}$.

We now describe our algorithm, sketchy-submodular-maximization (Algorithm 7). We first create $k$ independent linear sketches (recall that the process of creating a linear sketch for the input function is given as input to the algorithm). Then we run the following classical greedy submodular maximization algorithm with the modification that instead of directly evaluating the input function $f$ it queries the given sketch. Note that in each of the $k$ adaptive rounds, we use a different sketch. The classical greedy algorithm in each round simply looks at all subsets that have not been chosen and adds the one with the largest marginal gain to the output set (Nemhauser et al., 1978).

---

**Algorithm 7** sketchy-submodular-maximization

1: Initialize $\mathcal{C} \leftarrow \emptyset$.
2: **while** $|\mathcal{C}| \leq k$ **do**
3:    $\mathcal{C} \leftarrow \mathcal{C} \cup \operatorname{argmax}_{c \in d \setminus \mathcal{C}} g_{\boldsymbol{S}}(\{\boldsymbol{S} \cdot a_i\}_{i \in c|\mathcal{C}})$.
4: **end while**
5: Return $\mathcal{C}$.

---

We first analyze the memory usage. We are given that each sketch takes $O(s)$ space. Since there are $k$ rounds of adaptivity, the total space taken by the sketches is $O(sk)$. Both the update and reporting time will depend on the specific linear sketch.

Now, let us prove correctness. We assume by our theorem statement that our sketch $\boldsymbol{S}$ and corresponding function $g_{\boldsymbol{S}}$ give us a $(1 \pm \gamma)$-approximation to the queried values of our input function $f$. There are $k$ adaptive rounds. Since we create as many sketches and use a different one in each round, adaptivity between the rounds does not introduce error. In addition, despite getting $(1 \pm \gamma)$-approximations to all our queried values instead of the true queried values of our input function, we still get our desired approximation ratio by setting $\gamma = \epsilon/k$. This is proven and discussed in Theorem 5 of Horel & Singer (2016).

We also still get our approximation ratio with high probability. Since the error probability for each function evaluation is $O(1/(ndk))$, by a union bound over all $dk$ function evaluations, we have an error probability of at most $1 - 1/n$.

A.5    OMITTED PROOFS FROM SECTION 4 (COMPLEMENT OF $F_p$ LINEAR SKETCH)

**Claim A.2.** *k-Tuples-Sketch uses $\tilde{O}(\gamma^{-2})$ space and has an update time of $\tilde{O}(\gamma^{-2})$ and reporting time of $\tilde{O}(\gamma^{-2})$.*

*Proof.* We keep 3 $L_0$ sketches and $2t = \tilde{O}(\gamma^{-2})$ perfect $L_0$ samplers. Recall that $p$ is a constant. This proves the space usage and update time. The reporting time is dominated by computing the final output $n^p - \sum_{j=1}^{z'} (f_j')^p$ which takes $\tilde{O}(\gamma^{-2})$ time. Recall that we do not spend any time on values that have not been sampled. $\square$

Now we prove correctness. We first give the following result which we will use throughout the proof.

**Lemma A.3** (Lemma 3 of Bhattacharyya et al. (2016))**.** *Let $f_i$ and $\hat{f}_i$ be the frequencies of an item $i$ in a stream $\mathcal{S}$ (of length $n$) and in a random sample of $\mathcal{T}$ of size $r$ from $\mathcal{S}$, respectively. Then for*

$r \geq 2\gamma^{-2}\log(2\delta^{-1})$, *with probability* $1 - \delta$, *for every universe item $i$ simultaneously,*

$$\left|\frac{\hat{f}_i}{r} - \frac{f_i}{n}\right| \leq \gamma.$$

For the rest of the analysis, let us order the frequencies of the distinct values of vector $\mathbf{x}$ in non-increasing order as $f_1 \geq f_2 \geq \ldots \geq f_z$.

First note that in the algorithm we use $L_0^1$ and $\mathcal{S}$ to determine what value is $b$, or the value with frequency $f_1$. Then, if $f_b'$ is too small, then we simply output $n^p$. We now show that this is a good approximation.

**Claim A.4.** *If $f_1 \leq \gamma^{\frac{1}{p-1}} \cdot n$, then outputting $n^p$ is a $(1 \pm \gamma)$ approximation to $n^p - \sum_{i \in \mathcal{Z}} f_i^p$.*

*Proof.* Here, $\sum_{i \in \mathcal{Z}} f_i^p$ is greatest when there are $1/\gamma^{\frac{1}{p-1}}$ values each with true frequency $\gamma^{\frac{1}{p-1}} \cdot n$. So it is at most

$$\sum_i^{(\frac{1}{\gamma})^{\frac{1}{p-1}}} (\gamma^{\frac{1}{p-1}} \cdot n)^p = \left(\frac{1}{\gamma}\right)^{\frac{1}{p-1}} \cdot \gamma^{\frac{p}{p-1}} \cdot n^p = \gamma \cdot n^p.$$

Therefore, outputting $n^p$ is a $(1 \pm \gamma)$ relative approximation. $\qquad\square$

**Claim A.5.** *Using $L_0^1$ or $\mathcal{S}$ to estimate the frequency of a value $v$ outputs $f_v' = v \pm \frac{\gamma^{\frac{1}{p-1}}}{5 \cdot 2^p} \cdot n$.*

*Proof.* When using $\mathcal{S}$ (which is $t$ uniform samples of the nonzero entries of $\mathbf{x}$) to estimate the frequency of $v$, we find the frequency of $v$ among $\mathcal{S}$ and then scale by $w_2/t$. Here, $w_2$ is our $(1 \pm \varepsilon)$ with $\varepsilon = \frac{\gamma^{\frac{1}{p-1}}}{16 \cdot 2^p}$ estimate to the number of nonzero entries in $\mathbf{x}$.

By Lemma A.3 we incur at most $\varepsilon n = \frac{\gamma^{\frac{1}{p-1}}}{16 \cdot 2^p} \cdot n$ additive error from estimating the frequency of a value from $\mathcal{S}$ assuming that $w_2$ is exactly the number of nonzeros in $\mathbf{x}$. That combined with the error from estimating $\|\mathbf{x}\|_0$ with $w_2$ gives us at most $(2\varepsilon + \varepsilon^2) \cdot n \leq 3\varepsilon \cdot n \leq \frac{\gamma^{\frac{1}{p-1}}}{5 \cdot 2^p} \cdot n$ additive error. In addition, we use $L_0^1$ to determine the number of 0's to see if 0 is the value of the largest frequency. This incurs at most $\varepsilon \cdot n \leq \frac{\gamma^{\frac{1}{p-1}}}{5 \cdot 2^p} \cdot n$ error. $\qquad\square$

Recall that in the algorithm we output $n^p$ if $f_b'$ from $\mathcal{S}$ and $L_0^1$ is less than $\frac{3\gamma^{\frac{1}{p-1}}}{4} \cdot n$. Since we know the error in estimating each frequency is less than $\frac{\gamma^{\frac{1}{p-1}}}{4} \cdot n$ by Claim A.5, at worst all values had frequency $\gamma^{\frac{1}{p-1}} \cdot n$, and we output $n^p$. This does not incur too much error by Claim A.4.

In the rest of the analysis, we can assume that $f_1 \geq \frac{\gamma^{\frac{1}{p-1}}}{2} \cdot n$. We now claim that incurring $\gamma \cdot f_1^{p-1} \cdot (n - f_1)$ error still gives us the desired error guarantee.

**Claim A.6.** *Incurring $\gamma \cdot f_1^{p-1} \cdot (n - f_1)$ error gives us $\gamma \cdot (n^p - F_p)$ total error.*

*Proof.* We have that $n^p - F_p \geq f_1^{p-1} \cdot (n - f_1)$ for integers $p \geq 2$. $n^p - F_p$ counts the number of $p$-tuples (allowing repetitions from an individual item among the $n$) in which not all of the entries of the tuple have the same value. The right hand side counts $p$-tuples in which all but one entry are equal to the value of highest frequency (i.e. $f_1$) and the last has a different value.

Note that we can assume $p \leq \frac{\gamma}{2} \cdot n$ since $p$ is a constant. Therefore, we know that $f_1 \geq p$. $\qquad\square$

In all of the below, we assume the correctness of the $L_0$ sketches and perfect $L_0$ samplers. In the algorithm we have set their probability of error appropriately such that the probability of error across all of them is at most $5\delta/8$. In addition, by Lemma A.3, we have that using the $L_0$ samplers to estimate the frequencies as desired has error at most $2\delta/8$. So, we show that we incur at most error $\delta/8$ for the rest of the algorithm.

Recall that we estimate the frequency of the value of highest frequency differently if its estimated frequency is greater than $n/2$. Specifically, we instead subtract off the value from a $L_0$ sketch and query it. We will now show that estimating the frequency of this value does not incur too much error.

**Claim A.7.** *If $f_1 \geq \frac{2n}{3}$, the error incurred from our estimate of $f_1$ is at most $\frac{\gamma}{3} \cdot f_1^{p-1} \cdot (n - f_1)$.*

*Proof.* Let us denote the distinct value that has frequency $f_1$ in $\mathbf{x}$ as $b$. By Claim A.5, we incur at most $\varepsilon_1 = \frac{\gamma^{\frac{1}{p-1}}}{5 \cdot 2^p} \cdot n$ error in estimating the frequency of $b$ using $L_0^1$ and $\mathcal{S}$. Since $f_1 \geq \frac{2n}{3}$, the next largest frequency is at most $\frac{n}{3}$. Therefore, we will not mistake another value for $b$. In addition, we will find $b$ since in the algorithm we look for a estimated frequency greater than $\frac{n}{2}$.

Since we correctly identify $b$, then the following is true. In our algorithm we subtract off $b$ from $L_0^3$ (a linear $L_0$ sketch) and then query it to get $w_3$. Then we estimate the frequency as $n - w_3$. By the properties of the $L_0$ sketch, we incur at most $\Delta(f) = \varepsilon_2 \cdot (n - f_1)$ for $\varepsilon_2 = \frac{\gamma^{\frac{1}{p-1}}}{16 \cdot 2^p}$. Therefore, our total error is at most

$$(f_1 + \Delta(f))^p - f_1^p = \sum_{j=1}^{p} \left[ \binom{p}{j} f_1^{p-j} \Delta(f)^j \right] = \varepsilon \sum_{j=1}^{p} \left[ \binom{p}{j} f_1^{p-j} \Delta(f)^{j-1} \right]$$

$$\leq \Delta(f) \sum_{j=1}^{p} \left[ \binom{p}{j} f_1^{p-1} \right] = \Delta(f) f_1^{p-1} \cdot 2^p$$

giving us the desired error. Note that our estimate of $f_1$ could have been $f_1 - \Delta(f)$ but we have $|(f_1 + \Delta(f))^p - f_1^p| \geq |(f_1 - \Delta(f))^p - f_1^p|$. $\square$

Let us now consider the case where we do not have $f_1 \geq 2n/3$ but in the algorithm we identify a value $v$ with estimated frequency $f'_v \geq n/2$. By Claim A.5, we only incur $\varepsilon_1 = \frac{\gamma^{\frac{1}{p-1}}}{5 \cdot 2^p} \cdot n$ error in estimating the values using $\mathcal{S}$ and $L_0^1$ to identify the frequency of the highest frequency value. Therefore, it must be that $f_v \geq f_1 - 2\varepsilon_1 \cdot n$. So we incur total error $2\varepsilon_1 \cdot n + \varepsilon_2 \cdot (n - f_v) \leq 2\varepsilon_1 \cdot n + \varepsilon_2(n - f_1 + 2\varepsilon_1 \cdot n)$ in estimating this top frequency. However, we only estimate the frequency of $v$ using $L_0^3$ if $f' \geq n/2$, and we therefore know that $f_v = \Theta(f_1)$. Therefore we get total error $c \cdot (n - f_v)$ in estimating the frequency of $v$ and use similar analysis to Claim A.7 to get the desired error guarantee.

We now show that estimating the values of frequency at least $\frac{\gamma^{\frac{1}{p-1}}}{2} \cdot (n - f_1)$ does not incur too much error. We denote a set $\mathcal{F}$ which contains every value of $\mathbf{x}$ with frequency at least $\frac{\gamma^{\frac{1}{p-1}}}{2} \cdot (n - f_1)$.

**Claim A.8.** *The error incurred to the output from estimating $\sum_{i \in \mathcal{F}} f_i^p$ is at most $\frac{\gamma}{3} \cdot f_1^{p-1} \cdot (n - f_1)$ with probability at least $1 - \delta/8$.*

*Proof.* We first show how much error we incur by estimating the frequency of one value in $\mathcal{F}$. Take $\varepsilon = \frac{\gamma^{\frac{1}{p-1}}}{16 \cdot 2^p}$ We estimate the frequency $f_i$ for some $i$ (except $i = b$ if $f'_1 \geq n/2$) with

$$\frac{f_{i,t} + \varepsilon \cdot t}{t} \cdot ((n - f_1) + \varepsilon \cdot (n - f_1))$$

where $t$ is the number of uniform samples we take in the algorithm and $f_{i,t}$ is the frequency of value $i$ among the $t$ samples. The true answer is $f_{i,t} \cdot (n - f_1)/t$ so the error is at most

$$\frac{f_{i,t} \cdot \varepsilon \cdot (n - f_1) + \varepsilon \cdot t \cdot (n - f_1) + \varepsilon^2 \cdot t \cdot (n - f_1))}{t} \leq 2\varepsilon \cdot (n - f_1) + \varepsilon^2 \cdot (n - f_1) \leq 3\varepsilon \cdot (n - f_1).$$

By similar reasoning as above, choosing $b$ incorrectly and therefore subtracting off a different frequency to form $\mathcal{S}_*$ only increases this error by a constant factor. In addition, note that because in the algorithm we add all values with estimate frequency at least $\frac{\gamma^{\frac{1}{p-1}}}{4} \cdot (n - f'_1)$, we will put all values that are in $\mathcal{F}$ in $\mathcal{B}$ correctly. We now look at the error incurred in estimating all the frequencies of values in $\mathcal{B}$.

We denote $\Delta(f_i) = c \cdot \varepsilon \cdot (n - f_1)$ for some constant $c$. Let us consider all frequencies except $f_1$. We have that the error is at most

$$\sum_{i \in \mathcal{B}, i > 1} [(f_i')^p - f_i^p] = \sum_{i \in \mathcal{B}, i > 1} [(f_i + \Delta(f_i))^p - f_i^p]$$

$$= \sum_{i \in \mathcal{B}, i > 1} \left[ \sum_{j=1}^{p} \left( \binom{p}{j} f_i^{p-j} \Delta(f_i)^j \right) \right] \leq \sum_{i \in \mathcal{B}, i > 1} \left[ \Delta(f_i) \sum_{j=1}^{p} \left( \binom{p}{j} f_i^{p-1} \right) \right]$$

$$\leq \sum_{i \in \mathcal{B}, i > 1} \left[ \Delta(f_i) \cdot 2^p \cdot f_i^{p-1} \right] \leq 2^p \cdot f_1^{p-1} \cdot \sum_{i \in \mathcal{B}, i > 1} \Delta(f_i).$$

We will now show that $\sum_{i \in \mathcal{B}, i > 1} \Delta(f_i)$ is appropriately bounded. Note that $\sum_{i \in \mathcal{B}, i > 1} \Delta(f_i)$ is the sum of the errors in calculating the frequencies of values in $\mathcal{B}$ (except for $f_1$). When we estimate a frequency $f_i$ from $\mathcal{S}_*$ (which is made up of $t$ uniform samples), we are outputting the estimated frequency in our sample of size $t$ multiplied by $(n - f_1)/t$. Like before we can easily handle the error from calculating $(n - f_1)$. Therefore, we have $\mathbb{E}[f_i'] = f_i$ and $\mathrm{Var}[f_i'] \leq (n - f_1)/t \cdot f_i$. This gives us $\mathbb{E}\left[\sum_{i \neq 1} f_i'\right] = \sum f_i$ and $\mathrm{Var}[\sum_{i \neq 1} f_i'] \leq (n - f_1)/t \cdot \sum_{i \neq 1} f_i$. Recall that we have $\sum_{i \neq 1} f_i \leq n - f_1$. So, we can now apply Chebyshev's to get that with probability at least $1 - \delta/8$ we have $\sum_{i \in \mathcal{B}, i > 1} \Delta(f_i) \leq \frac{\varepsilon}{2} \cdot (n - f_1)$.

If we had $f_1 \leq 2n/3$, we get error $\Theta(\varepsilon n)$ from estimating its frequency from $\mathcal{S}$ and $L_0^1$ as proved by Claim A.5. Since we know that $\Theta(\varepsilon n) \leq f_1 \leq 2n/3$, by re-weighing $\varepsilon$ we get appropriate error. $\qquad \square$

We now deal with values $j$ such that $f_j \leq \frac{\gamma^{\frac{1}{p-1}}}{2} \cdot (n - f_1)$. We potentially do not approximate these frequencies. However, their contribution to $\sum f_i^p$ is low, and they give us small error as show below.

**Claim A.9.** *The error incurred by not estimating values with frequency less than $\frac{\gamma^{\frac{1}{p-1}}}{2} \cdot (n - f_1)$ is at most $\frac{\gamma}{3} \cdot (n^p - F_p)$.*

*Proof.* We first observe that we have $\sum_{i \neq 1} f_i = n - f_1$. So, $\sum_{i \notin \mathcal{F}} f_i^p$ is greatest when there are $\frac{2}{\gamma^{\frac{1}{p-1}}}$ coordinates of value $\frac{\gamma^{\frac{1}{p-1}}}{2} \cdot (n - f_1)$. So this sum (and therefore the error we incur) is at most

$$\sum_i^{2/\gamma^{\frac{1}{p-1}}} \left( \frac{\gamma^{\frac{1}{p-1}}}{2} \cdot (n - f_1) \right)^p \leq \frac{\gamma}{2^{p-1}} \cdot (n - f_1)^p.$$

We have that $(n - f_1)^p \leq n^p - f_1^p$ so we are getting $\frac{\gamma}{2^{p-1}} \cdot (n^p - f_1^p)$ total error.

The quantity that we want to estimate is $n^p - f_1^p - \sum_{i > 1} f_i^p$. We can see that

$$n^p - f_1^p - \sum_{i > 1} f_i^p \geq n^p - f_1^p - \frac{(n - f_1)^p}{c}$$

for some constant $c \geq 2$ since we have $\sum_{i > 1} f_1 = n - f_1$. Furthermore, we have that $n^p - f_1^p \geq (n - f_1)^p$. So, achieving $\frac{\gamma}{2^{p-1}} \cdot (n^p - f_1^p)$ gives us the desired error guarantee. $\qquad \square$

Therefore, combining all the claims above gives the result.

### A.6 PROOF OF THEOREM 4 (GENERAL FINGERPRINTING)

We now discuss our algorithm for general fingerprinting, general-fingerprinting-sketch (Algorithm 8) and prove Theorem 4. To utilize our general submodular maximization framework from Theorem 2, we need to provide a sketch that preserves queried values of the general fingerprinting function to within a $(1 \pm \gamma)$ factor. The general fingerprinting function receives as input a subset

of the columns of $\boldsymbol{A}$ and outputs how many pairs of users they separate. We can therefore see that maximizing this function gives us the desired output. Note that the general fingerprinting function is submodular since when adding a new column to a set $\mathcal{C}$ of columns, if this separates a pair of users that were previously not separated, then this column also separates that pair of users on some $T \subseteq \mathcal{C}$. It is also monotone since adding another column to $\mathcal{C}$ never decreases the function value.

---

**Algorithm 8** general-fingerprinting-sketch($n \times d$ matrix $\boldsymbol{A}$, $\varepsilon \in (0,1)$, $k \geq 0$)

---

1: $\gamma \leftarrow \varepsilon/k$.
2: **for** $j \in [d]$ **do**
3:     Maintain a $L_0$ sketch with error $\gamma$ and $\tilde{O}(\gamma^{-2})$ perfect $L_0$ samplers for the $j^{\text{th}}$ column of $\boldsymbol{A}$.
4: **end for**
5: **To answer a query:**
6: The query will ask for the function value on a subset of columns $\mathcal{C}$.
7: For each $j \in [d]$, view the $L_0$ samplers as a vector. We denote this vector as the "$L_0$ sampler sketch."
8: For all $j \in \mathcal{C}$, take the $L_0$ sketches and concatenate them into a matrix. Denote this as $L_1$.
9: For all $j \in \mathcal{C}$, take the $L_0$ sampler sketches and concatenate them into a matrix. Denote this as $L_2$.
10: Reduce the column dimension of $L_1$ and $L_2$ by right multiplying by a random vector $v$ from $\{-\text{poly}(ndk), \dots, \text{poly}(ndk)\}^{|\mathcal{C}|}$.
11: Run the sketch from Theorem 3 using $L_1$ and $L_2$ with $\delta = 1/(ndk)$, $\gamma = \varepsilon/k$, and $p = 2$ to estimate $\frac{n^2 - F_2}{2}$.

---

Let us analyze the memory usage. We keep one $L_0$ sketch per column of $\boldsymbol{A}$. As per Theorem 2, we must set $\gamma = \epsilon/k$ for our sketch. This makes the space of each $L_0$ sketch $\tilde{O}(k^2/\epsilon^2)$. So the total space for all $d$ columns is $\tilde{O}(dk^2/\epsilon^2)$. The space for each $L_0$ sampler is $\tilde{O}(\log^2 n)$, and we keep $\tilde{O}(dk^2/\varepsilon^2)$ of them giving us $\tilde{O}(dk^2/\epsilon^2)$. Using Theorem 2, our total space is therefore $\tilde{O}(dk^3/\epsilon^2)$. The update time is $\tilde{O}(dk^3/\varepsilon^2)$ since $k$ sketches will be created as in accordance with Theorem 2. The reporting time is also the same.

Now, we prove the correctness. As per our framework in Theorem 2, our result follows if we can show that our sketch provides $(1 \pm \gamma)$-approximations to all queried values to our general fingerprint function with probability $O(1/(ndk))$.

Upon a query to our function on a subset of columns $\mathcal{C}$, we return $g_S(\{S \cdot a_i\}_{i \in \mathcal{C}})$. To do this, for each type of sketch (both the $L_0$ sketch and the $L_0$-sampling sketch) for the columns of subset $\mathcal{C}$, we concatenate them and reduce them each to one column.

**Claim A.10.** *With probability $1 - 1/(ndk)$, for any rows $x$ and $y$ in $(\mathcal{S}\boldsymbol{A})_{\mathcal{C}}$ for sketch $\mathcal{S}$, they are distinct if and only if entry $x$ and $y$ of $[(\mathcal{S}\boldsymbol{A})_{\mathcal{C}}]\boldsymbol{v}$ are distinct for random vector $\mathbf{x}$ with entries in $\{-\text{poly}(ndk), \text{poly}(ndk)\}$.*

*Proof.* Let us look at two rows of $\boldsymbol{B} = (\mathcal{S}\boldsymbol{A})_{\mathcal{C}}$ that are distinct. We call these rows $\boldsymbol{B}_x$ and $\boldsymbol{B}_y$. Take $\boldsymbol{w}$ to be the vector that is formed from performing $\boldsymbol{B}_x - \boldsymbol{B}_y$. We first want to show that $\boldsymbol{w}^\intercal \boldsymbol{v} \neq 0$.

We have that $\boldsymbol{w}^\intercal \boldsymbol{v} = \boldsymbol{w}_1 \cdot \boldsymbol{v}_1 + \boldsymbol{w}_2 \cdot \boldsymbol{v}_2 + \cdots + \boldsymbol{w}_d \cdot \boldsymbol{v}_d$. Fixing the values of $\boldsymbol{v}_1$ through $\boldsymbol{v}_{d-1}$, there is only one value for $\boldsymbol{v}_d$ such that $\boldsymbol{w}^\intercal \boldsymbol{v} = 0$. Therefore, this "bad" event happens with probability at most $1/\text{poly}(ndk)$. Union bounding over all possible rows of $\boldsymbol{B}$, we have that with probability $1 - 1/(ndk)$ if rows $x$ and $y$ of $\boldsymbol{B}$ for any $x, y$ are distinct then entries $x$ and $y$ of $\boldsymbol{B}\boldsymbol{v}$ are distinct.

To finish up the proof, we want to show that if rows $x$ and $y$ of $\boldsymbol{B}$ for any $x, y$ are identical, then entries $x$ and $y$ of $\boldsymbol{B}\boldsymbol{v}$ are identical. This is clearly true with probability 1. $\qquad\square$

Now, we are in the vector case. We claim that the rest of the work is done by passing in $L_1$ and $L_2$ into our sketch from Theorem 3 with $p = 2$. For each distinct item $i$ in the vector, we denote its frequency as $f_i$. As we can see, $\binom{n}{2} - \sum_i \binom{f_i}{2} = \frac{n^2 - F_2}{2}$ is the general fingerprinting function. This is because $\binom{n}{2}$ denotes all pairs of users and by subtracting off $\sum_i \binom{f_i}{2}$ we are subtracting off pairs

of users that share identical values. Note the changes in the parameters of the input between here and in Theorem 3.

