# OpenReview forum: "Maximum Coverage in Turnstile Streams with Applications to Fingerprinting Measures"
_ICLR.cc/2025/Conference — Submitted to ICLR 2025_

### Official Review · Reviewer_8bWh · 2024-11-01

**Soundness:** 2
**Presentation:** 1
**Contribution:** 3
**Rating:** 5
**Confidence:** 3

**Summary:**

The paper provides streaming algorithms for maximum coverage and fingerprinting for risk measurement problems. The streaming algorithms are in the turnstile model (input elements can inserted or deleted). Previous results on streaming algorithms were known only for the insertion-only model. Let us discuss the main results for both problems:

Max-coverage:
In the maximum coverage problem, we are given subsets S1, ..., Sd of a universal set U (of size n) and an integer k, and the task is to find the k subsets that together cover the largest size subset of U. The problem is NP-hard. There is a poly-time (1-1/e)-approximation algorithm that is known to be tight. The input for this problem can be seen as a 0/1 matrix A of size nxd, where A(i, j)=1 iff i is in the subset S_j. In a previous work, McGregor and Vu (2018) gave a (1-1/e-\eps)-approximation streaming algorithm in the insertion-only set-arrival model using O(d/\eps^2) space. In the set-arrival model, the entire column of the matrix A is seen in one step. Bateni et al. (2017) gave a (1-1/e-\eps)-approximation algorithm in the insertion-only edge-arrival model using O(d/\eps^2) space. In the edge-arrival insertion-only model, a single matrix entry gets updated from 0 to 1. This paper explores the edge-arrival turnstile model, where in every step, one matrix entry may get updated from 0 to 1 or from 1 to 0.

Fingerprinting for Risk Management:
In targeted fingerprinting, the input is an n×d matrix A with n users and d features. The goal is to identify at most k features {f1,f2,...,fk} such that the number of users who share identical values at positions {f1,f2,...,fk} is minimized.

**Strengths:**

Extending the known results for the turnstile model is interesting.

**Weaknesses:**

There is significant scope for improving the write-up. There is a lack of clarity in many of the statements that leaves the reader confused:
- What is F_k in the abstract?
- The introduction assumes the knowledge about the definition of a sketch. Writing 1-2 sentences defining a sketch before using it in the discussion would be good.
- The abstract states the space usage to be O(d/\eps^2), but the main theorem (Theorem 1) gives the space-bound as O(d/\eps^3).
- Remark 1 is unclear and confusing. It starts talking about 'sampling rates', l_0-sampler, etc. without discussing any random process or algorithm. I had no option but to move on without understanding Remark 1.
- Lines (141-143): It is unclear what the estimation problem is. Are x_i values given in the streaming setting, or is it (i, \pm 1)? Unless this is made clear, I am not sure how to interpret Theorem 3.
- Understanding the sketch algorithm (Algorithm-1) is extremely challenging given that the format of the sketch is not defined. Is H_{\leq d} a matrix or a subset of (element, subset) pairs? How is this sketch updated in the stream on an insertion and deletion? This does not come out clearly from the pseudocode. In line 14 of Algorithm 1, it is said that "sketches and samplers handle updates." Which sketches are these?

**Questions:**

Some questions are mentioned in the other parts of the review.

---

> ### Author Response · Authors · 2024-11-28
>
> Thank you for your review and feedback. We answer your questions below.
>
> 1. Question: What is $F_k$ in the abstract?
>
> This is the $k^{th}$ frequency moment. We have clarified this in the abstract and preliminaries. In the revised version of the paper, we use the parameter $p$ instead of $k$ to avoid overlap with the input parameter $k$ for maximum coverage.
>
> 2. Question: The introduction assumes the knowledge about the definition of a sketch. Writing 1-2 sentences defining a sketch before using it in the discussion would be good.
>
> Thank you for the suggestion! We have included this in our revised uploaded paper.
>
> 3. Question: The abstract states the space usage to be $O(d/\varepsilon^2)$, but the main theorem (Theorem 1) gives the space-bound as $O(d/\varepsilon^3)$.
>
> Thank you for pointing out this typo. We have fixed it. The space bound is $O(d/\varepsilon^3)$.
>
> 4. Question: Remark 1 is unclear and confusing. It starts talking about 'sampling rates', l_0-sampler, etc. without discussing any random process or algorithm. I had no option but to move on without understanding Remark 1.
>
> We apologize for the confusing wording. We have removed Remark 1 from the paper.
>
> 5. Question: Lines (141-143): It is unclear what the estimation problem is. Are $x_i$ values given in the streaming setting, or is it $(i, \pm 1)$? Unless this is made clear, I am not sure how to interpret Theorem 3.
>
> We apologize for any confusion here and have provided more clarity before the theorem to specify this. We are getting updates of the form $(i, \pm 1)$.
>
> 6. Question: Understanding the sketch algorithm (Algorithm 1) is extremely challenging given that the format of the sketch is not defined. Is $H_{\leq d}$ a matrix or a subset of (element, subset) pairs? How is this sketch updated in the stream on an insertion and deletion? This does not come out clearly from the pseudocode. In line 14 of Algorithm 1, it is said that "sketches and samplers handle updates." Which sketches are these?
>
> We apologize for the confusing wording. We have improved the clarity of the algorithm description and pseudocode of Section 3. Specifically, we first describe our algorithm in a non-streaming and non-low space setting, where we assume direct access to the matrix $A$ in its final state (i.e., no further updates). Here, we prove the algorithm's correctness. We then proceed step by step to demonstrate how to implement this algorithm as a linear sketch that accommodates updates while detailing what exactly is stored in memory at each step.
>
> In regards to your specific question, the algorithm proceeds as follows. We first subsample rows of the input matrix $A$ to form a smaller submatrix $A’$. Importantly, we do not explicitly store these subsampled rows that form $A’$. Rather, we use a hash function to keep track of whether a row has been subsampled to form $A’$. This hash function uses only logarithmic bits of space. Then, whenever an update comes, the run that considers submatrix $A’$ only considers updates to $A’$ (rather than all updates).
>
> Now, consider $A’$. Each row of $A’$ is hashed to a bucket. Again, these rows are not stored explicitly as doing so would require too much space. Instead, we keep a hash function to keep track of which bucket a row has been hashed to. Consider an individual bucket. Suppose that the rows hashed here are those corresponding to items 1, 4, and 5. Then, in this bucket, we will only consider updates to entries that are in the 1st, 4th, or 5th rows. Now, we think of vector $v$ in the following way. Concatenate rows 1,4, and 5 such that the first entries of $v$ are those of row $1$, the next entries are of row $4$, and the last entries are of row $5$. We again do not store $v$ explicitly but instead store $L_0$ samplers for $v$. Each $L_0$ sampler uniformly at random holds a nonzero entry of $v$. These samplers dynamically update to ensure uniformity as updates occur.  This is what we mean when we say that samplers and sketches handle updates (since all the samplers and sketches we use are linear sketches themselves). The nonzero entries that the $L_0$ samplers hold will form $A_*$ (previously named $H_{\leq d}$). So, $A_*$ is a subset of the nonzero entries of the original input matrix $A$.
>
> Thank you again for your comments and questions, and please let us know if there is anything else we can clarify.

---

> > ### Author Response · Authors · 2024-12-02
> > **Follow up to Reviewer 8bWh**
> >
> > Dear Reviewer 8bWh,
> >
> > Thank you again for your review.
> >
> > We believe we have thoroughly addressed your questions about the writing. If any of your questions concerns have not been addressed, could you please let us know before the end of the discussion phase?
> >
> > Many thanks, The authors

---

### Official Review · Reviewer_PurF · 2024-11-03

**Soundness:** 3
**Presentation:** 3
**Contribution:** 3
**Rating:** 6
**Confidence:** 2

**Summary:**

The paper addresses the problem of constructing a linear sketch for the maximum $k$ coverage problem. Given a set of $n$ items and $d$ subsets, the objective of the maximum $k$ coverage problem is to select $k$ subsets that maximize the number of items covered. The problem can be represented using a matrix $A \in \{0, 1\}^{n \times d}$, and the goal of the linear sketch is to find a matrix $S$ such that $SA$ is significantly smaller than $A$ while still enabling an approximate solution to the original $k$ coverage problem using $SA$.
Since the sketch is linear, it naturally extends to the turnstile model, where the entries of $A$ can be updated over time.

The paper also demonstrates the application of this sketching technique to the problem of fingerprinting for risk management, with empirical studies indicating substantial speed improvements over previous methods.

**Strengths:**

1. The paper constructs a linear sketch that supports input updates for the maximum $k$ coverage problem.
2. Experimental evaluations demonstrate a significant speedup compared to prior work.
3. Overall, I find the paper to be well-written.

**Weaknesses:**

1.	Some explanation is missing for the algorithm. E.g., the paper claims that algorithm 1 construct a linear sketch. Does this imply that the $L_0$ sampler used in Algorithm 1 is also a linear sketch? The same question applies to the $L_1$ sketch. It would be helpful to explicitly clarify whether these sketches are linear, and if they are, to provide brief explanations or references that detail how they function as linear sketches. This additional context would aid readers in understanding the overall linearity of Algorithm 1.

**Questions:**

1.	Line 209, $|x_i| \ge \epsilon^2 || x ||_p$. What is the value of $p$, or does this apply to arbitrary value of $p$?
2.	Line 222, $\epsilon$ is missing in the $\tilde{O}$ notation, whereas in line 224, the $\epsilon$ is not omitted in the $\tilde{O}$ notation.
3.	Line 226, $\alpha - \epsilon$ -- > $1 - 1 / e - \epsilon$.

---

> ### Author Response · Authors · 2024-11-28
>
> Thank you for your review and feedback. We address your questions and comments below.
>
> Comment/Question: “Some explanation is missing for the algorithm. E.g., the paper claims that algorithm 1 construct a linear sketch. Does this imply that the L_0 sampler used in Algorithm 1 is also a linear sketch? The same question applies to the L_0 sketch. It would be helpful to explicitly clarify whether these sketches are linear, and if they are, to provide brief explanations or references that detail how they function as linear sketches. This additional context would aid readers in understanding the overall linearity of Algorithm 1.”
>
> Thank you for pointing this out. Yes, all the samplers and sketches we use in this paper are linear sketches themselves. We have clarified this in the preliminaries in the revised version of the paper. We have also expanded in the preliminary section of the paper the section on linear sketches. In addition, we have added additional description and clarity to Section 3 for our max coverage algorithm.  Specifically, we first describe our algorithm in a non-streaming and non-low space setting, where we assume direct access to the matrix $A$ in its final state (i.e., no further updates). Here, we prove the algorithm's correctness. We then proceed step by step to demonstrate how to implement this algorithm as a linear sketch that accommodates updates while detailing what is stored at each step.
>
> We now address your numbered questions.
> 1. This is the definition of an $\ell_p$ heavy hitter, which is defined for $p \geq 0$.
> 2. Thank you for pointing this out. We will fix that typo.
> 3. We have fixed this typo.
>
> Thank you again for your comments and questions, and please let us know if there is anything else we can clarify.

---

> > ### Author Response · Authors · 2024-12-02
> > **Follow up to Reviewer PurF**
> >
> > Dear Reviewer PurF,
> >
> > Thank you again for your review.
> >
> > We believe we have addressed your questions about the algorithm. If any of your concerns have not been addressed, could you please let us know before the end of the discussion phase?
> >
> > Many thanks, The authors

---

### Official Review · Reviewer_Yept · 2024-11-03

**Soundness:** 2
**Presentation:** 2
**Contribution:** 2
**Rating:** 5
**Confidence:** 3

**Summary:**

This work studies the maximum coverage problem in a streaming setting:  Given $d$ sets over an universe $[n]$ and an integer $k$. Find $k$ sets whose union is maximized.  The input (represented as $n \times d$ matrix) arrives as a stream. Earlier works studied this problem in the insertion-only streaming model. This work studies the problem in the turnstile model, where deletions are allowed. The main contribution is the design of a sketch-based algorithm that use $\tilde {O} (d/\epsilon^3)$ space.

**Strengths:**

1. This is the first algorithm for the maximum coverage problem in the turnstile model.

**Weaknesses:**

1. The writing could be improved to enhance the readability of the paper. I am not able to completely understand the proposed algorithm and verify the claims. Please see the Questions for details.
2.  There is a large body of work on streaming submodular maximization. A discussion on the relationships of those works to the current work is missing.  For example,
      [1]. https://proceedings.neurips.cc/paper_files/paper/2020/file/6fbd841e2e4b2938351a4f9b68f12e6b-Paper.pdf
      [2]. https://dl.acm.org/doi/10.1145/3519935.3519951
      [3]. https://proceedings.neurips.cc/paper_files/paper/2020/file/9715d04413f296eaf3c30c47cec3daa6-Paper.pdf
3. Experimental results are not evaluated on turnstile streams.

**Questions:**

1. Is the model is strict-turnstle model?
2. Line 200: L_1 sketches are trivial. Do you mean L_0 Skteches?
3. The description of the algorithm is confusing and not precise.  For example
    a. Line 9. All rows are concatenated to obtain v. However, each entry in the matrix is 0-1. So $v$ is a binary vector? I assume $v$ contains row numbers/elements of the universe? (For example, if elements $i$ is in sets 3, 4 8. Then the vector $v$ contains the number $i$, at 3 positions.
    b. When you are keeping an L_0 sample of $v$, what do they contain?  It seems to me that they contain a sample of rows  (excluding all zero rows, and hashed into the same bucket) from $A'_m$?  Is this correct?
    c. Line 23: I do not understand what it means to "if $r$ has less then ..... edges among $L_0$ samplers". Clarifying the above question will help understand this line.

4. Claim 3.1: Consider the instance where $m = 1$. $A'_1$ contains approximately half the rows. Shouldn't this need $n/2$ memory? What am I Missing?
5. Line 276-277: $k \log d/\epsilon^2$ is a fixed number. How can this be OPT?
6. Claim 3.2: McGregor & Vu's proof relies on (set) insertion only model? Is it easy to see that it translates into a turnstile model?
7. Line 294: What are $c_1, c_2, \cdots$?  They are not defined earlier
8.  As defined a linear sketch is a matrix drawn from a family of matrices. The algorithm is implicitly defining a family of matrices. Can you define these matrices more explicitly? For the sketch to be linear, the $L_0$ sampler needs to be linear. This should be clarified.

I will revise my score after the discussion period.

---

> ### Author Response · Authors · 2024-11-28
>
> Thank you for your review and feedback. We respond to your questions and comments below.
>
> We apologize for any unclear wording or details. Below, we address your questions directly. Additionally, we have revised the pseudocode and expanded the explanation of the algorithm in Section 3 (maximum coverage) in the revised version of the paper to provide greater clarity.
>
> We also thank you for highlighting these works ([1], [2], and [3]). We note that the referenced studies pertain to the dynamic model, which differs from the streaming model. In the dynamic model, the primary focus is on time complexity, whereas in the streaming model, the emphasis is on achieving sublinear space (though our algorithms also attain sublinear update time). In the revised version of the paper, we have added a brief discussion in the introduction to clarify the distinctions between the dynamic and streaming models.
>
> While we do not run our experiments on a turnstile stream, our primary goal for the proposed linear sketch was to show significant runtime improvements with small memory. We therefore demonstrated these improvements in our experiments all while keeping good comparative accuracy to previous work which used linear memory and time. Since our algorithm is a linear sketch, it will work on any ordering of the entries. In addition, it will accommodate any ordering of updates to the matrix it is applied to.
>
> We now respond to your questions.
> 1) We do not require the model to be strict-turnstile. Updates take the form $(i, j, +1)$ or $(i, j, -1)$, meaning we either add $1$ or subtract $1$ from entry $(i,j)$ of the input matrix $A$. In addition, our algorithm can support entries of $A$ being both negative and positive. In maximum coverage, we take any non-zero value (including a negative value) at entry $(i,j)$ to mean that item $i$ is covered by subset $j$.
>
> 2) We apologize for the typo, and you are correct that we use $L_0$ sketches. To improve clarity, we have revised both the pseudocode and the proof in Section 4 in the revised version.
>
> 3) We apologize for any unclear wording. In the revised Section 3, we have improved clarity by reorganizing the explanation. We first describe our algorithm in a non-streaming and non-low space setting, where we assume direct access to the matrix $A$ in its final state (i.e., no further updates). Here, we prove the algorithm's correctness. We then proceed step by step to demonstrate how to implement this algorithm as a linear sketch that accommodates updates where we detail what is stored in each step.
>
> In response to your question, for maximum coverage, we interpret the input matrix $A$ as consisting of nonzeros and 0’s. If entry $(i,j)$ is $0$ in $A$, that means that $j$ does not cover item $i$. If entry $(i,j)$ is $1$ in $A$, that means that $j$ does cover $i$. Note that you can indeed think of any nonzero as a 1 for this purpose. We include nonzeros to make our algorithm more general and also because it is useful for our targeted fingerprinting application.
>
> The algorithm proceeds as follows. We first subsample rows of the input matrix $A$ to form a smaller submatrix $A’$. Importantly, we do not explicitly store these subsampled rows that form $A’$. Rather, we use a hash function to keep track of whether a row has been subsampled to form $A’$. This hash function uses only logarithmic bits of space. Then, whenever an update comes, the run that considers submatrix $A’$ only considers updates to $A’$ (rather than all updates).
>
> Now, consider $A’$. Each row of $A’$ is hashed to a bucket. Again, these rows are not stored explicitly as doing so would require too much space. Instead, we keep a hash function to keep track of which bucket a row has been hashed to. Consider an individual bucket. Suppose that the rows hashed here are those corresponding to items 1, 4, and 5. Then, in this bucket, we will only consider updates to entries that are in the 1st, 4th, or 5th rows. Now, we think of vector $v$ in the following way. Concatenate rows 1,4, and 5 such that the first entries of $v$ are those of row $1$, the next entries are of row $4$, and the last entries are of row $5$. We again do not store $v$ explicitly but instead store $L_0$ samplers for $v$. Each $L_0$ sampler uniformly at random holds a nonzero entry of $v$. These samplers dynamically update to ensure uniformity as updates occur.
>
> In Line 23, we examine all the nonzero entries stored in the $L_0$ samplers that we stored. To construct the final sketch, our goal is to retain ~$d/k$ nonzero entries per row within the subsampled universe $A’$. However, some rows may have started off with fewer nonzero entries (or had none at all). This if statement is accounting for this possibility.
>
> 4) This was answered in the response to question 3.
>
> We continue our responses in the next official comment.

---

> ### Author Response · Authors · 2024-11-28
>
> 5) Here $k \log d/\varepsilon^2$ is not OPT in the original input matrix $A$. The first step involves subsampling rows from matrix $A$ to form $A’$ such that the correct sampling rate corresponds to the $A’$ that has OPT value of $k \log d / \varepsilon^2$.
>
> 6) It is true that the algorithm of McGregor & Vu relies on the set insertion only model. However, the only part of the proof we use from their paper is the result that we can subsample rows from the matrix $A$ to form submatrix $A’$ such that OPT in $A’$ is $k \log d/\varepsilon^2$, and that solving the problem on $A’$ results in at most an $\varepsilon$ factor loss in our approximation. This is a fairly standard proof which shows that when OPT is at this rate, the number of items in the union of any $k$ subsets is preserved appropriately.
>
> 7) $c_1, c_2, \ldots$ are defined within the proof. The process we describe can be understood as follows. Start by considering the set of large items we have. Next, select a subset that covers $c_1$ items, then remove those items and the subset itself. After that, pick another subset that covers $c_1$ items, and repeat this process for $k$ iterations. We use this to prove an upper bound on the number of large items. We use a similar proof to prove an upper bound on the number of edges among small items.
>
> 8) You are correct that we are implicitly defining a family of matrices. This is because storing the sketching matrix explicitly would require too much space. Recall that the sketching matrix must have $n$ columns. Therefore, we describe the implicit representation and talk about how updates are performed. You are correct that L_0 samplers need to be linear. They are linear sketches themselves, and we have clarified this in the preliminaries in the revised version of the paper.
>
> Thank you again for your thorough comments, and please let us know if there is anything else we can clarify.

---

> > ### Author Response · Authors · 2024-12-02
> > **Follow up to Reviewer Yept**
> >
> > Dear Reviewer Yept,
> >
> > Thank you again for your review.
> >
> > We believe we have thoroughly addressed your questions about the writing, relationship to existing work, and nature of our experiments. If any of your concerns have not been addressed, could you please let us know before the end of the discussion phase?
> >
> > Many thanks, The authors

---

### Official Review · Reviewer_N3fx · 2024-11-04

**Soundness:** 2
**Presentation:** 1
**Contribution:** 2
**Rating:** 5
**Confidence:** 4

**Summary:**

In this paper the authors consider the problem of choosing at most k subsets from a stream such that the number of distinct items covered by the subsets is maximized. This is an interesting problem with applications in other areas - as demonstrated in the paper. The authors give a O~(d/\epsilon^2) algorithm where d is the number of sets in the stream and \epsilon is an approximation parameter.

**Strengths:**

The algorithms presented in the paper are interesting.

**Weaknesses:**

It is clear that the most important parameter is k. And when analysis the complexity of the algorithms the authors have avoided discussing the dependency of the space complexity on k. This makes it very hard to properly judge the true contribution of the paper.

**Questions:**

What is the actual dependency on k?

---

> ### Author Response · Authors · 2024-11-28
>
> Thank you for your review and feedback. In our space analysis, the dependence on $k$ is poly$(\log k)$. Additionally, since the space already includes poly$(\log d)$ factors, and we can assume $k \leq d$ (as $k > d$ allows us to directly output all input subsets), any polylogarithmic factors in $k$ factors are absorbed.
>
> Therefore, when suppressing log factors, we achieve a space complexity of O(d/eps^3) space, consistent with prior work but applicable in a more general model. We apologize for any unclear wording, and we have provided additional clarity on this in the revised version of the paper in the "Our Contributions" section.
>
> Thank you again for your comments and questions, and please let us know if there is anything else we can clarify.

---

> > ### Author Response · Authors · 2024-12-02
> > **Follow up to Reviewer N3fx**
> >
> > Dear Reviewer N3fx,
> >
> > Thank you again for your review.
> >
> > We believe we have addressed your question about the dependency on k. If any of your questions or concerns have not been addressed, could you please let us know before the end of the discussion phase?
> >
> > Many thanks, The authors

---

### Meta-Review · Area_Chair_KXUk · 2024-12-15

**Metareview:**

The paper presents a streaming algorithm for max-k coverage problem which is a classical NP-Hard problem. Many such algorithms are already known, and the paper claims to present the first algorithm in the turnstile model. However, the paper writing is troublesome and as pointed out by the reviewers, the authors failed to do a thorough comparison of prior works. Given this problem has been extensively studied already in theoretical computer science, and due to above weaknesses, I cannot recommend the paper for acceptance in its current stage.

**Additional Comments On Reviewer Discussion:**

Rebuttals were ok but did not convince the reviewers to change their scores.

---

### Decision · Program_Chairs · 2025-01-22

Reject